# IL-21-mediated reversal of NK cell exhaustion facilitates anti-tumour immunity in MHC class I-deficient tumours

Hyungseok Seo[1], Insu Jeon[1], Byung-Seok Kim[2], Myunghwan Park[1], Eun-Ah Bae[1], Boyeong Song[1], Choong-Hyun Koh[3], Kwang-Soo Shin[3], Il-Kyu Kim[1,3], Kiyoung Choi[4], Taegwon Oh[4], Jiyoun Min[5], Byung Soh Min[6], Yoon Dae Han[6], Suk-Jo Kang[5], Sang Joon Shin[7], Yeonseok Chung[2] & Chang-Yuil Kang[1,3,4]

During cancer immunoediting, loss of major histocompatibility complex class I (MHC-I) in neoplasm contributes to the evasion of tumours from host immune system. Recent studies have demonstrated that most natural killer (NK) cells that are found in advanced cancers are defective, releasing the malignant MHC-I-deficient tumours from NK-cell-dependent immune control. Here, we show that a natural killer T (NKT)-cell-ligand-loaded tumour-antigen expressing antigen-presenting cell (APC)-based vaccine effectively eradicates these advanced tumours. During this process, we find that the co-expression of Tim-3 and PD-1 marks functionally exhausted NK cells in advanced tumours and that MHC-I downregulation in tumours is closely associated with the induction of NK-cell exhaustion in both tumour-bearing mice and cancer patients. Furthermore, the recovery of NK-cell function by IL-21 is critical for the anti-tumour effects of the vaccine against advanced tumours. These results reveal the process involved in the induction of NK-cell dysfunction in advanced cancers and provide a guidance for the development of strategies for cancer immunotherapy.

[1] Laboratory of Immunology, Department of Molecular Medicine and Biopharmaceutical Sciences, Graduate School of Convergence Science and Technology, Seoul National University, Seoul 08826, Republic of Korea. [2] Laboratory of Immune Regulation, Research Institute of Pharmaceutical Science, College of Pharmacy, Seoul National University, Seoul 08826, Republic of Korea. [3] Laboratory of Immunology, Research Institute of Pharmaceutical Science, College of Pharmacy Seoul National University, Seoul 08826, Republic of Korea. [4] Cellid, Inc., Seoul 08826, Republic of Korea. [5] Department of Biological Science, Korea Advanced Institute of Science and Technology, Daejeon 34141, Republic of Korea. [6] Department of Surgery, Yonsei University College of Medicine, Seoul 03722, Republic of Korea. [7] Department of Internal Medicine, Yonsei University College of Medicine, Seoul 03722, Republic of Korea. Correspondence and requests for materials should be addressed to C.-Y.K. (email: cykang@snu.ac.kr).

A lthough a number of anti-cancer immunotherapies are currently being investigated in clinical trials, one of the major obstacles in treating advanced cancer is that tumour cells escape host immune responses via the downregulation of major histocompatibility complex class I (MHC-I)[1,2]. The malignant transformation and subsequent selection of highly metastatic cells by the immune system result in the loss of MHC class I in the neoplasm, contributing to tumour evasion from immunosurveillance by cytotoxic T lymphocytes. In addition, the downregulation of MHC class I

**Figure 1 | MHC class I downregulation affects anti-tumour immunity.** (**a**) Each group of mice ($n = 6$) was vaccinated with the indicated cellular vaccine at day 6 after TC-1 tumour s.c. injection ($1 \times 10^5$) at day 0. (**b**) Each group of mice ($n = 8$) was vaccinated with the indicated cellular vaccine at day 10, 24 and 41 after TC-1 tumour s.c. injection ($5 \times 10^5$) at day 0. The tumour size was measured three times weekly. (**c,d**) Each group of mice ($n = 10$) was vaccinated with the indicated cellular vaccine at day 7, 14 and 21 after TC-1 tumour i.v. injection ($1 \times 10^5$) at day 0. (**d**) *In vivo* imaging of TC-1 metastatic mice (day 40) treated with metalloproteinase. (**e**) Expression of H-2K$^b$ and H-2D$^b$ on tumours. C57BL/6 mice were implanted in the flank with $1 \times 10^5$ TC-1 cells (day 0 indicates *in vitro* cultured tumour cells). (**f**) H-2K$^b$ and H-2D$^b$ expression (**e**) is shown with tumour size. (**g,h**) Each graph depicts the specific lysis of extracted TC-1 cells at different time points. CD8$^+$ T cells and NK cells were isolated from the spleens of B/Mo/E6E7/αGC-immunized mice at days 1 and 7 as effector cells. (**i**) The data presented in **g–h** reassembled with the H-2K$^b$ and H-2D$^b$ level on tumours and the percentage of specific lysis when the effector-to-target ratio was 30:1. The data in **a** were analysed by two-way analysis of variance (ANOVA) with Bonferroni multiple comparison tests. The data in **c** were analysed using a log-rank (Mantel–Cox) test (conservative) *$P < 0.05$, **$P < 0.01$, ***$P < 0.001$, ****$P < 0.0001$. The data are representative of three independent experiments that included three to ten mice per group. All values represent the mean ± s.e.m.

in tumours induces natural killer (NK)-cell dysfunction, leading to the outgrowth of MHC class I-deficient tumours[3,4]. However, the underlying mechanisms involved in the induction of NK-cell dysfunction by MHC class I-deficient tumour cells and the best way to overcome the tolerogenic tumour microenvironment in advanced cancer remain to be elucidated[5].

Co-inhibitory receptors, such as programmed death 1 (PD-1) and T-cell immunoglobulin and mucin domain 3 (Tim-3), play a crucial role in mediating T-cell exhaustion in both viral infections and tumours[6,7]. The expression of these receptors has been identified in diverse immune cell populations including T cells, B cells and myeloid cells. Although previous studies demonstrated that the PD-1/PD-L1 and Tim-3/ligands of Tim-3 signalling down-modulated the cytotoxicity of NK cells against tumour cells[8,9], their expression on NK cells was not well documented until a few recent human studies reported PD-1 and Tim-3 expression on NK cells of cancer patients[10,11]. Nevertheless, the roles of these inhibitory receptors in the anti-cancer effector functions of NK cells remain elusive.

The IL-21 receptor (IL-21R) is expressed on NK, B, T and dendritic cells[12]. Several studies have reported that IL-21 acts directly on viral antigen-specific CD8$^+$ T cells to enhance their functional responses and to limit exhaustion during chronic viral infection[13–15]. IL-21 promotes the maturation of NK cell progenitors and activates the anti-tumour effects of NK cells through the NKG2D pathway[16,17]. In addition, IL-21 activates cytotoxic programs in both CD8$^+$ T and NK cells, thus providing potent cytotoxic effector arms against cancer cells[18]. Based on these studies, several clinical trials are currently underway[19].

We have previously reported that an invariant natural killer T (NKT) cell ligand, alpha-galactosylceramide (αGC), loaded on a tumour antigen (tAg)-expressing B cell- and monocyte-based vaccine (B/Mo/tAg/αGC) elicited diverse anti-tumour immune responses[20–22]. In this study, we found that B/Mo/tAg/αGC effectively eradicated otherwise resistant MHC class I-deficient tumour cells by activating NKT cells and inducing tumour antigen-specific cytotoxic T-cell responses. Whereas MHC class I-deficient tumour cells selectively induced Tim-3$^+$PD-1$^+$ NK cells with impaired cytotoxicity in the tumour microenvironment, B/Mo/tAg/αGC vaccination restored the cytotoxic capacity of NK cells. In addition, we found that the functional recovery of exhausted Tim-3$^+$PD-1$^+$ NK cells by vaccination was solely dependent on the activation of PI3K-AKT-Foxo1 and STAT1 signalling pathways by IL-21 produced by NKT cells. Accordingly, the addition of recombinant IL-21 restored the function of intratumoural Tim-3$^+$PD-1$^+$ NK cells both in animal models and in human cancer patients.

## Results

**Effects of the vaccine for advanced tumours.** To investigate whether B/Mo/tAg/αGC has anti-tumour effects on large established tumours, we first developed a B/Mo/tAg/αGC vaccine expressing the E6/E7 tumour Ag of human papillomavirus-associated cancer (B/Mo/E6E7/αGC). We found that B/Mo/E6E7/αGC elicited activation of NKT (Supplementary Fig. 1A) and NK cells (Supplementary Fig. 1B) and induced antigen-specific CTL responses (Supplementary Fig. 1C). A single vaccination on day 7 with B/Mo/E6E7/αGC was successful for the treatment of mice bearing small E6/E7-expressing TC-1 tumours (Fig. 1a) and protected mice against tumour re-growth (Supplementary Fig. 2). Multiple vaccinations at late time points effectively eradicated large established TC-1 tumours (Fig. 1b), and lung metastases derived from TC-1 tumour cells were efficiently eradicated by vaccination with B/Mo/E6E7/αGC (Fig. 1c,d). To investigate the effects of B/Mo/E6E7/αGC on MHC class I-downregulated

tumours, we analysed the kinetics of H-2K$^b$ and H-2D$^b$ expression on tumours. A gradual downregulation of H-2K$^b$ and H-2D$^b$ was observed with tumour progression (Fig. 1e,f). The cytotoxic activity of NK cells gradually increased during tumour progression, while the cytotoxicity of CD8$^+$ T cells gradually decreased (Fig. 1g,h). The cytotoxicity of NK cells was inversely correlated with the expression of H-2K$^b$ and H-2D$^b$ on tumours, whereas that of CD8$^+$ T cells was positively correlated with the expression of H-2K$^b$ and H-2D$^b$ (Fig. 1i). This observation prompted us to address whether MHC class I-deficiency can affect NK cell functions[23]. To this end, we established H-2K$^b$ and H-2D$^b$ KO or β2-microglobulin (β$_2$m)-deficient tumours (Supplementary Fig. 3A,B). The progression of tumour growth in the respective syngeneic mice was comparable between WT and MHC class I-deficient tumours (Supplementary Fig. 3C–E).

Several mouse and human studies have shown that NKT-cell activation triggers anti-tumour immune responses by enhancing NK cell activation[24,25]. To verify that this was also the case with MHC class I-deficient tumours, mice were injected with αGC-loaded B cells and monocytes (B/Mo/αGC) followed by MHC class I-deficient tumour inoculation. The B/Mo/αGC administration resulted in a reduction in the growth of MHC class I-deficient tumours compared with B/Mo (Fig. 2a–c). We also found that treatment with anti-NK1.1 or anti-asialo GM1 significantly reversed the inhibition of tumour growth, which indicated that the anti-tumour immunity was dependent on NK and NKT cells (Supplementary Fig. 4 and Fig. 2d–f). However, human malignant primary tumours are well known for their heterogeneous MHC class I expression levels[2,26]. Therefore, we hypothesized that NKT-cell-ligand-loaded tumour antigen-expressing vaccination can eradicate both MHC class I-sufficient and MHC class I-deficient tumour cells by inducing adaptive and innate immune responses. MHC class I-sufficient tumours could be eradicated by adaptive immune responses, such as CD8$^+$ T cells, and MHC class I-deficient tumours could be eradicated by innate immune responses, such as the response of NK cells. To address this hypothesis, we established a heterogeneous tumour model in which a mixture of MHC class I sufficient- and MHC class I-deficient tumour cells was inoculated at different ratios. The data revealed that our vaccine regimen induced complete tumour eradication and a long-lasting cure in 90% of the mice bearing heterogeneous tumours and thus contain up to 10% MHC class I-deficient TC-1 tumour cells (Fig. 2g,h). B/Mo/E6E7/αGC administration induced robust tumour eradication and durable cures in >70% of the mice bearing heterogeneous tumours, which contained 20% MHC class I-deficient TC-1 tumour cells. In contrast, B/Mo/E6E7 or B/Mo/ αGC administration elicited weaker anti-tumour responses and reduced survival rates (Fig. 2i). Antibody depletion experiments showed that CD8$^+$ T and NK cells were crucial for heterogenic tumour eradication and led to reductions in the survival rates (Fig. 2j). These results indicate that our vaccine regimen is effective against the heterogeneous tumours by inducing both adaptive and innate immune responses.

**MHC-I-deficient tumours induce Tim-3 and PD-1 on NK cells.** Although recent studies have demonstrated that MHC class I-downregulation or deficiency in tumour cells induces NK cell anergy, NKT-cell-ligand-loaded antigen-presenting cell (APC)-based tumour Ag-expressing vaccination efficiently induced the eradication of tumours containing MHC class I-deficient cells. From this perspective, we hypothesized that NK anergy in mice bearing MHC class I-deficient tumours might have been restored by the vaccination. To investigate this possibility, we analysed the cell surface marker expression pattern and cytotoxic function of

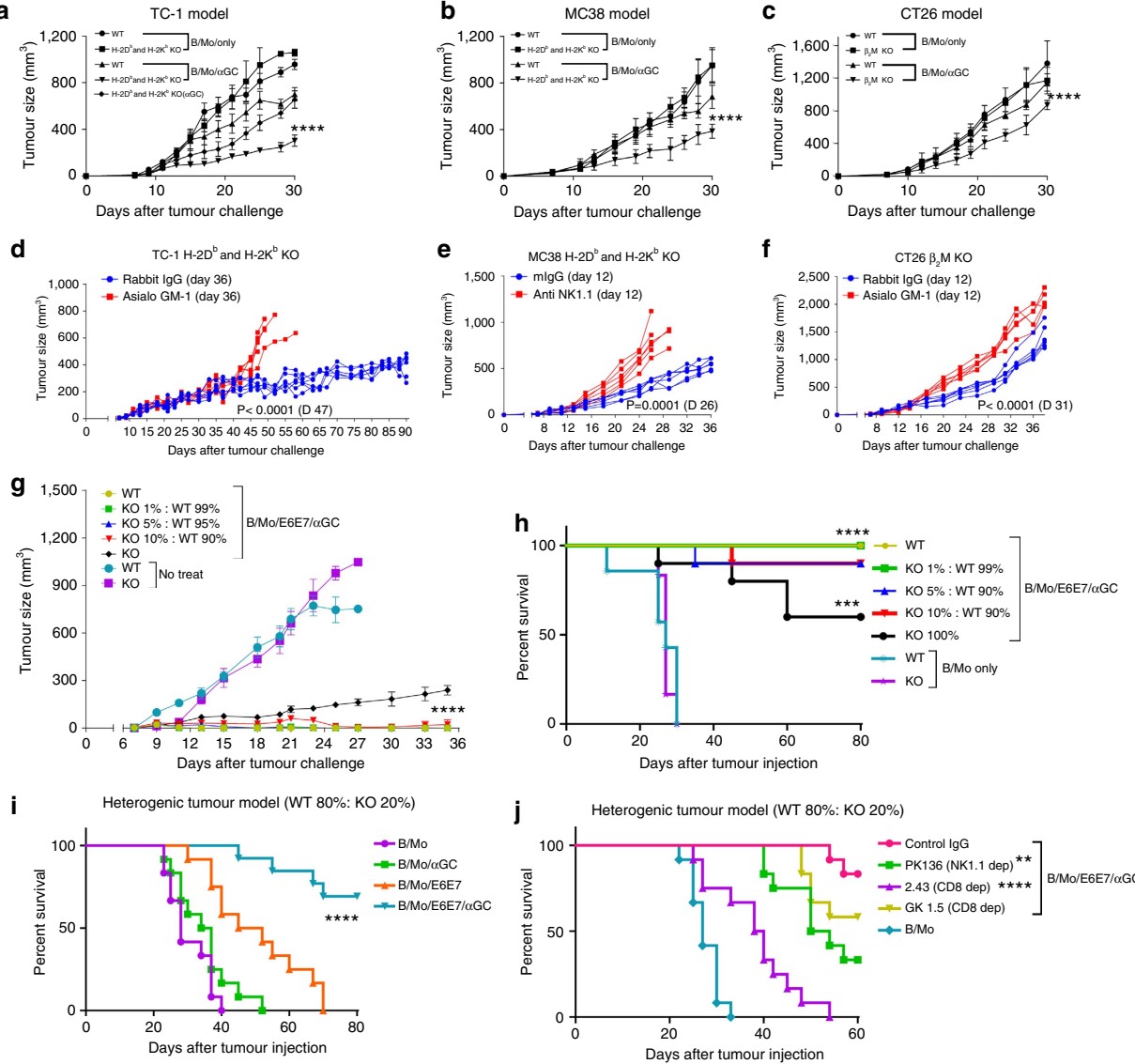

**Figure 2 | Eradication of heterogeneous tumours by induced adaptive and innate immune responses.** (**a–c**) Subcutaneous growth of WT and MHC class I knockout (KO) tumour cells (TC-1, MC38: H-2K$^b$ and H-2D$^b$ KO, CT26: β2m KO) in each group of mice ($n = 6$) treated with B/Mo ($1 \times 10^6$), B/Mo/αGC ($1 \times 10^6$) and αGC ($1 \mu g$) every 7 days. (**d–f**) Subcutaneous growth of the indicated tumour cells in each group of mice ($n = 6$) treated with B/Mo ($1 \times 10^6$) or B/Mo/αGC ($1 \times 10^6$) every week. The tumour-bearing mice were treated with control IgG, anti-NK1.1 (PK136) or anti-asialo GM-1 to deplete NK cells on day 36 (TC-1) or day 12 (MC38 and CT26). (**g,h**) Each group of mice ($n = 10$) was vaccinated with the indicated cellular vaccine every 7 days after s.c. injection of the indicated mixed (WT and MHC class I-deficient tumour) or solid (TC-1) tumours ($1 \times 10^5$) at day 0. (**i,j**) Survival over time with depleting antibodies for indicated cells (PK136 for NK cells depletion, 2.43 for CD8$^+$ T-cell depletion and GK1.5 for CD4$^+$ T-cell depletion) administered i.p.1 day before initiation of the vaccination. Each group of mice ($n = 12$) was vaccinated with the indicated cellular vaccine every 7 days after s.c. injection of the mixed (TC-1 WT and MHC class I-deficient tumour ratio = 80:20) tumours ($1 \times 10^5$) at day 0. **a–g** were analysed by a two-way ANOVA with Bonferroni multiple comparison tests. The data in **h–j** were analysed by a log-rank (Mantel–Cox) test (conservative) *$P < 0.05$, **$P < 0.01$, ***$P < 0.001$, ****$P < 0.0001$. The data are representative of three independent experiments that included six to ten mice per group. All values represent the mean ± s.e.m.

NK cells in MHC class I-deficient tumour microenvironments. Recent studies have suggested that Tim-3 expression is a marker of NK-cell activation or exhaustion[9,10]. We observed that Tim-3 expression on tumour-infiltrating NK cells was dramatically accelerated in mice bearing H-2K$^b$ and H-2D$^b$ KO tumours compared with WT tumour-bearing mice, and most Tim-3$^+$ NK cells co-expressed PD-1 (Fig. 3a). The levels of Tim-3 and PD-1 expression on NK cells in the spleen were very low, regardless of the MHC class I expression on tumours, but those in tumour-draining lymph nodes (TdLNs) were slightly increased in mice bearing H-2K$^b$ and H-2D$^b$ KO tumours compared with WT tumour-bearing mice, although this increase was not

statistically significant (Supplementary Fig. 5). The expression levels of NK-cell-activating receptors, including Ly49D, DNAM-1, CD69 and CD160, were significantly higher in Tim-3$^+$ PD-1$^+$ NK cells compared with Tim-3$^-$ PD-1$^-$ cells. The expression of NK-ell-inhibitory receptors such as NKG2A/C/E, KLRG1, TIGIT and 2B4 was comparable between the groups, but Ly49A was downregulated in Tim-3$^+$PD-1$^+$ NK cells (Fig. 3b). To investigate whether the tumours could directly induce the expression of Tim-3 and PD-1 on NK cells, we co-cultured purified NK cells with tumour cells. When NK cells were co-cultured with WT tumour cells, Tim-3 was slightly induced starting on day 1, and PD-1 was not induced until day 5.

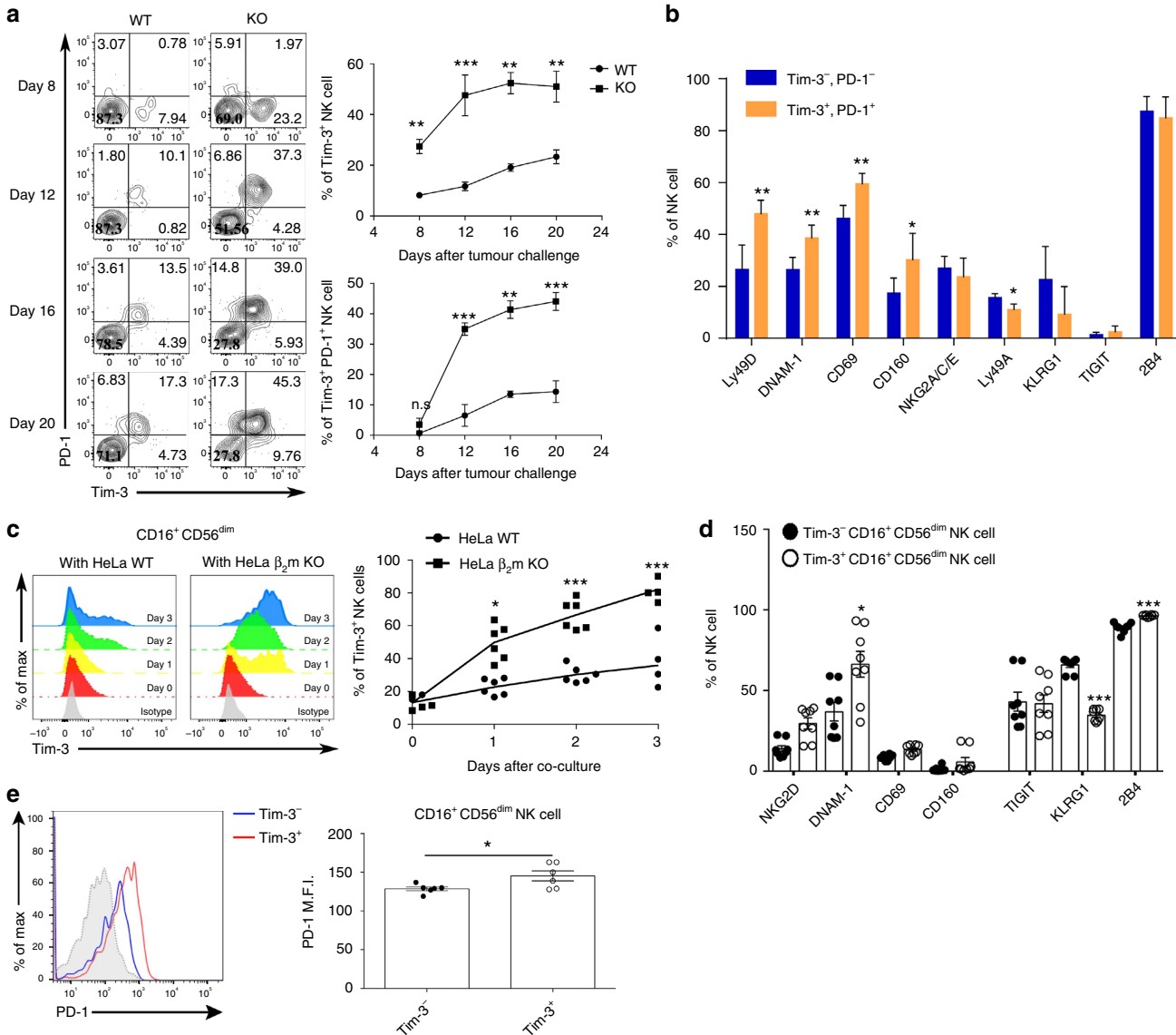

**Figure 3 | Natural killer cells in MHC class I-deficient tumours exhibit accelerated Tim-3 and PD-1 expression.** (**a**) The kinetic expression levels of Tim-3 and PD-1 on tumour-infiltrating natural killer cells from TC-1 WT or H-2K$^b$ and H-2D$^b$ KO tumour-bearing mice ($n=5$) were analysed by flow cytometry. (**b**) Tim-3$^+$PD-1$^+$ or Tim-3$^-$PD-1$^-$ NK cells from TC-1 H-2K$^b$ and H-2D$^b$ KO tumours were stained with antibodies for the activating receptors Ly49D, DNAM-1, CD69 and CD160 or the inhibitory receptors NKG2A/C/E, Ly49A, KLRG1, TIGIT and 2b4. (**c**) The Tim-3 expression level on human NK cells was analysed after CD16$^+$CD56$^{dim}$ human NK cells ($n=6$) were co-cultured with HeLa WT or β2m KO cells. (**d**) Tim-3$^+$ or Tim-3$^-$ human NK cells ($n=8$) co-cultured with HeLa β2m KO cells for 3 days were stained with antibodies for the activating receptors NKG2D, DNAM-1, CD69 and CD160 or the inhibitory receptors TIGIT, KLRG1,2B4 and PD-1 (**e**). The data in **a,c,d,e** were analysed by a two-tailed unpaired Student's $t$-test. *$P<0.05$, **$P<0.01$, ***$P<0.001$. The data are representative of three independent experiments that included three to five mice per group and six to eight human PBMC samples. All values represent the mean ± s.e.m.

However, when NK cells were co-cultured with MHC class I-deficient tumour cells, Tim-3 was highly induced starting on day 1, and PD-1 was induced on day 5 (Supplementary Fig. 6A). To analyse whether soluble factors produced by tumours could induce Tim-3 and PD-1, NK cells were both co-cultured with MHC class I-deficient tumour cells using a transwell system and in tumour-conditioned medium. The expression of Tim-3 and PD-1 on NK cells was induced when the NK cells and MHC class I-deficient tumours were directly co-cultured (Supplementary Fig. 6B). In human NK cells, Tim-3 expression was rapidly upregulated upon co-culture with β2m-deficient HeLa cells compared with NK cells co-cultured with WT HeLa cells (Fig. 3c). Human NK-cell-activating receptors, including NKG2D, DNAM-1 and CD69, and the inhibitory receptor 2B4

were significantly upregulated on Tim-3$^+$ NK cells compared to Tim-3$^-$ cells. However, expression of the inhibitory receptor KLRG1 was significantly downregulated in Tim-3$^+$ NK cells (Fig. 3d). The levels of PD-1 on Tim-3$^+$ NK cells were slightly higher than those on Tim-3$^-$ cells (Fig. 3e). These results collectively suggest that Tim-3$^+$PD-1$^+$ NK cells express higher levels of activating receptors.

**Tim-3$^+$ PD-1$^+$ NK cells are functionally exhausted in tumour.** Although Tim-3 and PD-1 were initially suggested to be activation markers of T cells, recent studies have demonstrated that Tim-3 and PD-1 co-expression is a marker of exhausted T cells after repeated TCR stimulation during chronic infection or in the

cancer microenvironment[7,27]. To investigate whether Tim-3$^+$ PD-1$^+$ NK cells induced by MHC class I-deficient tumours represent activated or exhausted NK cells, we analysed the effector functions of NK-cell populations isolated from mice bearing WT or H-2K$^b$ and H-2D$^b$ KO tumours. We determined that the frequency of IFN-$\gamma^+$ NK cells was remarkably decreased in tumour-infiltrating lymphocytes (TILs) and TdLNs of mice bearing H-2K$^b$ and H-2D$^b$ KO tumours compared with those bearing WT tumours (Supplementary Fig. 7C), although the frequency and number of NK cells were comparable between the two groups (Supplementary Fig. 7A,B). ERK1/2 kinase is a crucial factor for the induction of cytokine production and cytotoxicity in NK cells[28]. The levels of phospho-ERK1/2 were significantly reduced in TIL NK cells from MHC class I-deficient tumour-bearing mice compared with TIL NK cells from WT tumour-bearing mice (Supplementary Fig. 8A).

To investigate the interrelationship between Tim-3 and PD-1 expression and function on NK cells, we analysed the kinetics of the expression of these receptors and the IFN-$\gamma$ secretion of intratumoural NK cells from mice bearing WT and H-2K$^b$ and H-2D$^b$ KO tumours. In mice bearing WT tumours, the expression of Tim-3 and PD-1 on intratumoural NK cells reached 50% at 28 days after tumour injection. In contrast, Tim-3 and PD-1 were expressed much more rapidly on intratumoural NK cells from mice bearing H-2K$^b$ and H-2D$^b$ KO tumours than on intratumoural NK cells from mice bearing WT tumours, reaching up to 70% at 28 days after tumour injection. We also observed that IFN-$\gamma$ production of intratumoural NK cells from both groups of mice sharply decreased in response to the increased expression of Tim-3 and PD-1 (Fig. 4a). Of note, among the NK cells isolated from MHC class I-deficient tumours, Tim-3$^+$ PD-1$^+$ NK cells exhibited a dramatically reduced expression of effector molecules such as IFN-$\gamma$, CD107a, granzyme B and perforin compared with their Tim-3$^-$ PD-1$^-$ counterparts (Fig. 4b). These results suggest that when NK cells are stimulated, Tim-3 and PD-1 are first expressed on NK cells as an activation signature. However, after persistent stimulation in the absence of MHC class I, Tim-3$^+$ PD-1$^+$ expression represented the exhausted NK-cell population. In addition, T-bet expression on Tim-3$^+$PD-1$^+$ NK cells in H-2K$^b$ and H-2D$^b$ KO tumours was also reduced compared with Tim-3$^-$ PD-1$^-$ NK cells, whereas the expression of Eomes was somewhat increased on Tim-3$^+$PD-1$^+$ NK cells (Fig. 4c). Moreover, ERK1/2 phosphorylation was significantly decreased in Tim-3$^+$PD-1$^+$ NK cells (Fig. 4d). Consistent with their exhausted phenotype, the cytotoxicity of the Tim-3$^+$PD-1$^+$ NK cells against H-2K$^b$ and H-2D$^b$ KO TC-1 tumours was much lower than the cytotoxicity exhibited by Tim-3$^-$ PD-1$^-$ NK cells (Fig. 4e). We also found that Tim-3$^+$PD-1$^+$ intratumoural NK cells in WT tumour-bearing mice had a similar exhausted phenotype, in accordance with the spontaneous downregulation of MHC class I on WT tumour cells (Supplementary Fig. 8B).

To determine whether the PD-1/PD-L1 axis transmits negative signals on NK cells, we examined whether the tumour cells expressed PD-L1. Our data showed that MC38 tumour cells had a high expression level of the PD-L1 molecule on their surface (Supplementary Fig. 9A). To examine the effect of anti-PD-1 antibody on exhausted NK cells, we treated MC38 H-2K$^b$ and H-2D$^b$ KO tumour-bearing mice, which were depleted of CD8$^+$ T cells by 2.43 mAb, with the anti-PD-1 antibody. This treatment significantly suppressed tumour growth in mice bearing MC38 H-2K$^b$ and H-2D$^b$ KO tumours (Supplementary Fig. 9B). Collectively, our data suggest that the PD-1/PD-L1 axis transmits negative signals on NK cells and that blocking the PD-1 signal could restore the anti-tumour effects of exhausted NK cells. When human NK cells were co-cultured with $\beta_2$m-deficient HeLa cells, the IFN-$\gamma$ and granzyme B levels in Tim-3$^+$ NK cells were

gradually decreased after restimulation with the NKG2D ligand ULBP-2 (Supplementary Fig. 10A). A gradual reduction in T-bet expression in Tim-3$^+$NK cells was highly correlated with the pattern of IFN-$\gamma$ production, whereas the expression pattern of Eomes was opposite that of T-bet (Supplementary Fig. 10B). Altogether, these data suggest that repeated stimulation of NK cells by MHC class I-deficient tumours induces the functional exhaustion of NK cells marked by the co-expression of Tim-3 and PD-1 in mice and humans.

**IL-21 reverses the functions of exhausted NK cells.** Given that vaccination with B/Mo/$\alpha$GC effectively controls the growth of MHC class I-deficient tumours in an NK- and NKT-cell-dependent manner, we hypothesized that the vaccination could revive the otherwise defective effector functions of NK cells in mice bearing MHC class I-deficient tumours (Fig. 2a–f). Thus, we investigated whether B/Mo/$\alpha$GC could reverse the exhausted status of NK cells. The frequency of IFN-$\gamma$-producing NK cells was dramatically increased with B/Mo/$\alpha$GC vaccination compared with B/Mo alone, regardless of the MHC class I expression level in the tumours (Supplementary Fig. 11A). The increase in IFN-$\gamma$-producing NK cells upon B/Mo/$\alpha$GC vaccination was dependent on NKT cells because J$\alpha$281-deficient mice failed to demonstrate a similar increase (Supplementary Fig. 11B). NKT cells are known to secrete various cytokines, including IFN-$\gamma$, IL-4, TNF-$\alpha$ and IL-21 (refs 29,30). When B/Mo/$\alpha$GC was injected, we confirmed that the NKT cells produced IFN-$\gamma$, IL4, TNF-$\alpha$ and IL-21 (Fig. 5a, Supplementary Fig. 12A). Therefore, we next sought to determine whether NKT-cell-produced cytokines were responsible for the B/Mo/$\alpha$GC-mediated recovery of NK cell functions in H-2K$^b$ and H-2D$^b$ KO tumours. Because IL-21 and IFN-$\gamma$ are well-known stimulators of NK cell functions[19,31], we first tested whether these cytokines could revive the effector functions of exhausted NK cells in tumours. The addition of IFN-$\gamma$ partially retrieved the IFN-$\gamma$-producing capacity of exhausted NK cells, whereas the addition of IL-21 completely restored the functions of NK cells isolated from H-2K$^b$ and H-2D$^b$ KO tumours to levels comparable to those of NK cells isolated from WT tumours (Fig. 5b). We also found that ERK activation in NK cells was significantly increased by both IFN-$\gamma$ and IL-21 in vitro (Fig. 5c). However, the addition of IL-4 or TNF-$\alpha$ did not retrieve the IFN-$\gamma$-producing capacity of exhausted NK cells (Supplementary Fig. 12B).

To confirm the IFN-$\gamma$- and IL-21-dependent reversal of NK cell functions in vivo, we administered an anti-IFN-$\gamma$-neutralizing Ab or anti-IL-21R-blocking Ab after B/Mo/$\alpha$GC vaccination. The B/Mo/$\alpha$GC-mediated recovery of IFN-$\gamma$ production and CD107a degranulation in Tim-3$^+$PD-1$^+$ NK cells was significantly diminished in mice treated with the anti-IL-21R blocking Ab, while partial inhibition was observed in mice treated with the anti-IFN-$\gamma$ Ab (Fig. 5d). Analysis of perforin and granzyme B production also confirmed that the functional reversal of Tim-3$^+$PD-1$^+$ NK cells by B/Mo/$\alpha$GC was primarily IL-21-dependent (Supplementary Fig. 13). The recovery of T-bet expression in Tim-3$^+$PD-1$^+$ NK cells following B/Mo/$\alpha$GC vaccination was also completely abrogated by IL-21R blockade (Fig. 5e). To address whether alterations in effector cytokine production and T-bet expression could lead to changes in the cytotoxicity exhibited by each NK-cell subset, we tested the cytotoxicity of purified Tim-3$^-$ PD-1$^-$ and Tim-3$^+$PD-1$^+$ NK cells isolated after B/Mo/$\alpha$GC vaccination. We found that B/Mo/$\alpha$GC vaccination restored the cytotoxic functions of Tim-3$^+$PD-1$^+$ NK cells in MHC class I-deficient tumours and that IL-21 was required for this process because the IL-21R blockade significantly abrogated the recovery of NK cell cytotoxicity (Fig. 5f).

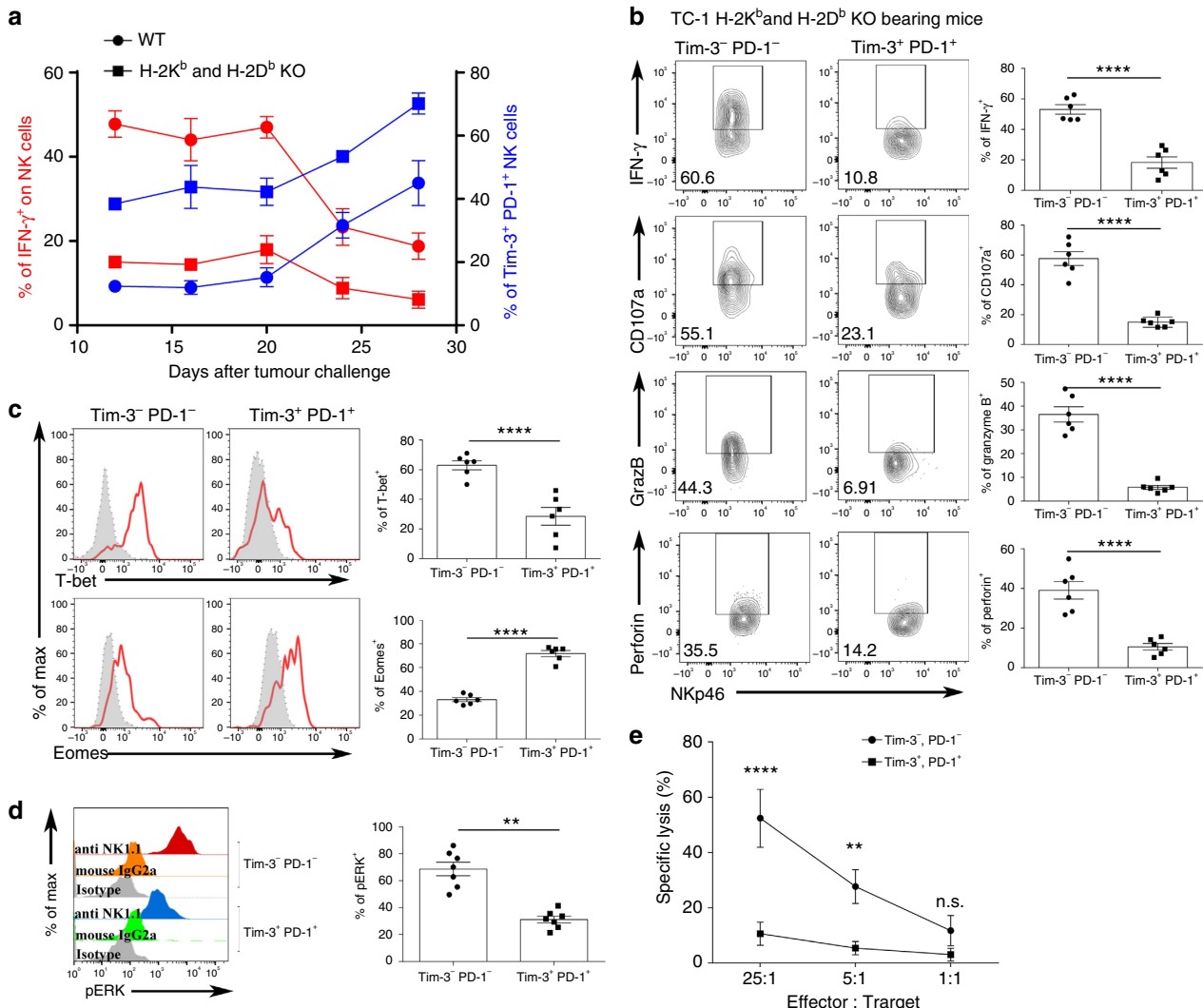

**Figure 4 | Infiltrating natural killer cells in MHC class I-deficient tumours are functionally exhausted.** (**a**) The expression of Tim-3, PD-1 and IFN-γ on intratumoural NK cells from mice bearing MC38 WT or H-2K$^b$ and H-2D$^b$ KO tumours was measured over time ($n = 5$). Intratumoural NK cells were additionally restimulated with anti-NK1.1 for the analysis of IFN-γ. (**b–e**) Twelve days after TC-1 H-2K$^b$ and H-2D$^b$ KO implantation, infiltrating lymphocytes were restimulated *in vitro* with anti-NK1.1, and the (**b**) IFN-γ, CD107a, granzyme B and perforin production, (**c**) T-bet and Eomes expression, and **d** ERK phosphorylation were determined in Tim-3$^+$PD-1$^+$ or Tim-3$^-$PD-1$^-$ natural killer cells by flow cytometry. (**e**) Intratumoural Tim-3$^+$PD-1$^+$ and Tim-3$^-$ PD-1$^-$ NK cells were isolated from TC-1 H-2K$^b$ and H-2D$^b$ KO-bearing mice (pooled from 20 mice). They were co-cultured with $^{51}$Cr labelled TC-1 H-2K$^b$ and H-2D$^b$ KO cells as target cells. The data in **a–d** were analysed by a two-tailed unpaired Student's *t*-test. The data in **e** were analysed by a two-way ANOVA with Bonferroni multiple comparison tests *$P < 0.05$, **$P < 0.01$, ***$P < 0.001$, ****$P < 0.0001$, n.s. = not significant. The data are representative of three independent experiments. All values represent the mean. ± s.e.m.

Consistent with these results, the regression of H-2K$^b$ and H-2D$^b$ KO TC-1 tumour growth following B/Mo/αGC vaccination was partially abrogated by IFN-γ neutralization, while IL-21R blockade almost completely restored tumour progression to the same extent observed in unvaccinated mice (Fig. 5g). This pattern was reproducible in two different MHC class I-deficient tumour models (Fig. 5h,i).

Next, we sought to determine whether IL-21 could directly induce the functional recovery of exhausted NK cells in MHC class I-deficient tumours. To achieve this goal, we employed Rag1$^{-/-}$ mice to eliminate the effects of IL-21 on T cells. When we intratumourally injected recombinant IL-21 (rIL-21) into MC38 H-2K$^b$ and H-2D$^b$ KO tumour-bearing Rag1$^{-/-}$ mice, tumour growth was significantly inhibited with an increase in tumour-infiltrating NK cells (Supplementary Fig. 14A,B). Although the levels of Tim-3 and PD-1 expression were decreased, the expression of IFN-γ and CD107a was increased

by intratumoural injection of rIL-21 (Supplementary Fig. 14C–E). To establish and isolate fully exhausted NK cells, we used an IFN-γ-eYFP-reporter (GREAT) mouse system. We isolated Tim-3$^+$PD-1$^+$eYFP$^-$ NK cells from MC38 H-2K$^b$ and H-2D$^b$ KO tumour-bearing GREAT mice and stimulated them with anti-NK1.1 in the presence of rIL-21. We observed that IFN-γ secretion of rIL-21-treated Tim-3$^+$PD-1$^+$eYFP$^-$ NK cells was significantly increased compared with that of vehicle-treated Tim-3$^+$PD-1$^+$eYFP$^-$ NK cells (Fig. 5j). To analyse the direct anti-tumour effects of rIL-21-treated Tim-3$^+$PD-1$^+$eYFP$^-$ NK cells, these cells were adoptively transferred into IL-2Rγ$^{-/-}$ Rag2$^{-/-}$ mice (lacking B cells, T cells and NK cells) bearing MC38 H-2K$^b$ and H-2D$^b$ KO tumours. We observed a significant inhibition of tumour growth in mice that received rIL-21-treated Tim-3$^+$PD-1$^+$eYFP$^-$ NK cells. The growth inhibition in the mice that received these cells was slightly higher, although not statistically significant, than that in mice that received Tim-

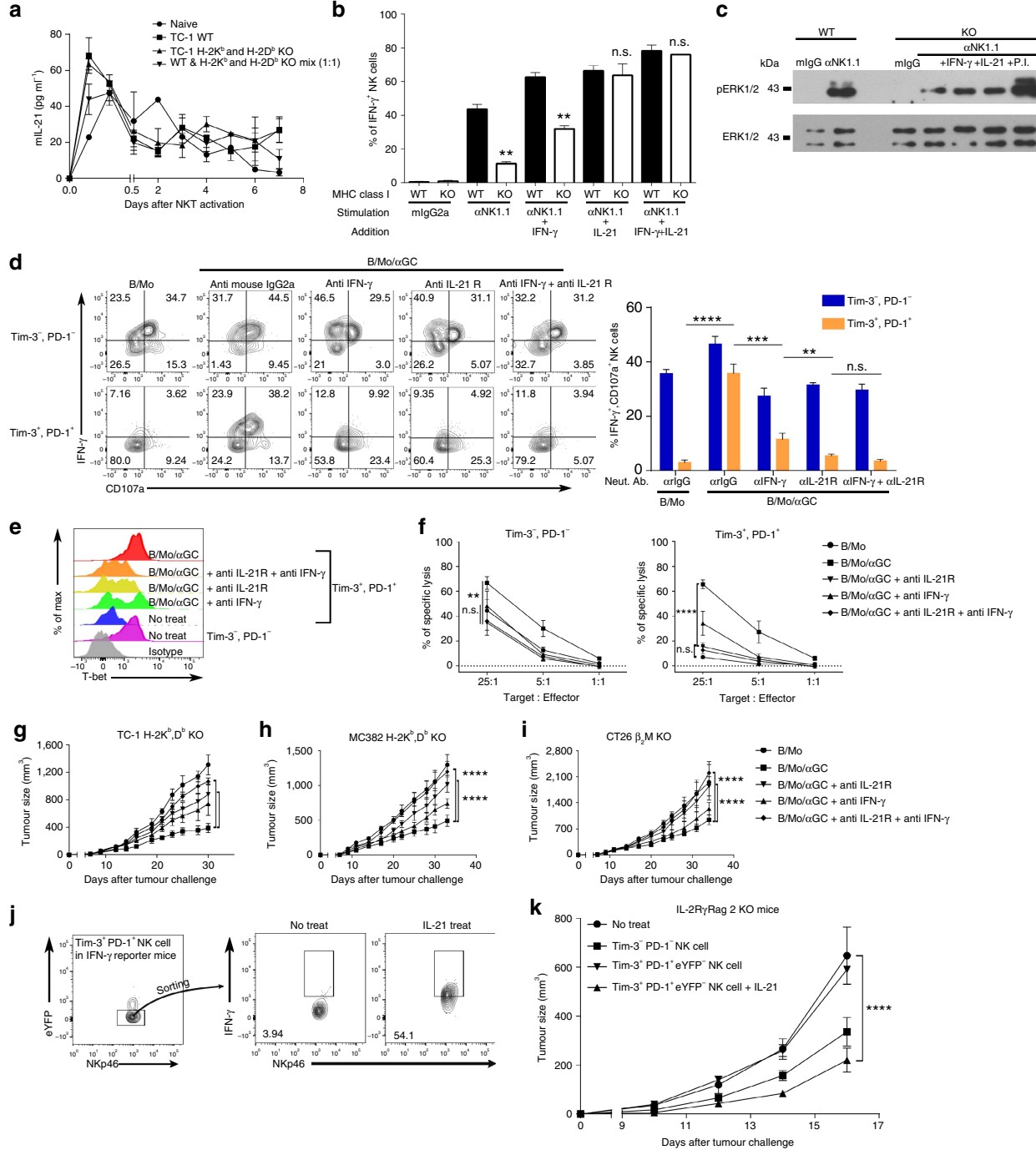

**Figure 5 | IL-21-dependent functional reversal of exhausted Tim-3[+] PD-1[+] NK cells.** (**a**) The cytokine profile in the serum of B/Mo/αGC-injected indicated tumour-bearing mice at various times. The levels of IL-21 were detected by ELISA. (**b,c**) Twelve days after TC-1 WT or H-2K[b] and H-2D[b] KO implantation, intratumoural lymphocytes were stimulated *in vitro* with IFN-γ (20 ng ml[−1]) and IL-21 (20 ng ml[−1]) alone or in combination and after they were restimulated with anti-NK1.1. (**b**) The IFN-γ production was determined by intracellular staining, (**c**) and ERK phosphorylation was determined by western blot analysis. (**d–g**) Nine days after TC-1 H-2K[b] and H-2D[b] KO implantation, anti-IFN-γ (500 μg per mouse) or anti-IL-21R (300 μg per mouse) or both were injected via i.p. and 2 days later, B/Mo/αGC (1 × 10[6] cells per mouse) was injected along with the neutralizing antibodies. In addition, 12 h later, (**d,e**) infiltrating lymphocyte were restimulated *in vitro* with anti-NK1.1 and (**d**) IFN-γ, CD107a, (**e**) T-bet expression were assessed by flow cytometry. (**f**) Infiltrating Tim-3[+]PD-1[+] and Tim-3[−]PD-1[−] NK cell were isolated in TC-1 H-2K[b] and H-2D[b] KO-bearing mice (pooled from 20 mice). They were co-cultured with [51]Cr labelled TC-1 H-2K[b] and H-2D[b] KO. (**g–i**) Subcutaneous growth of TC-1 (**g**), MC38 (**h**) and CT26 (**i**) WT and MHC Class I KO tumour cells in each mice (*n* = 6) treated with B/Mo(1 × 10[6]), B/Mo/αGC(1 × 10[6]) every week with indicated neutralizing antibodies. (**j**) IFN-γ-eYFP-reporter (GREAT) mice were implanted with MC38 H-2K[b] and H-2D[b] KO tumour cells and intratumoural Tim-3[+] PD-1[+]eYFP[−] NK cells were sorted and incubated overnight in the presence or absence IL-21 (20 ng ml[−1]) with anti-NK1.1 stimulation last 5 h. IFN-γ secretion was analysed by flow cytometry. (**k**) Tumour growth in IL-2Rγ[−/−]Rag2[−/−] mice that received the indicated group of cells (1 × 10[5]) (*n* = 10). MC38 H-2K[b] and H-2D[b] KO tumour cells (5 × 10[4]) were inoculated 1 day before NK cell transfer. The data in **a,b,d** were analysed by a two-tailed unpaired Student's *t*-test. The data in **f–k** were analysed by a two-way ANOVA with Bonferroni multiple comparison tests. *$P < 0.05$, **$P < 0.01$, ***$P < 0.001$, ****$P < 0.0001$, ns = not significant. The data are cumulative from at least two independent experiments. All values represent the mean ± s.e.m.

3⁻PD-1⁻ NK cells (Fig. 5k). Collectively, our results suggest that IL-21 could restore the function of exhausted Tim-3⁺PD-1⁺ NK cells and effectively inhibit tumour progression.

**IL-21 mediated mechanism of reversal of exhausted NK cells.** To further explore the underlying signalling mechanisms of IL-21-mediated functional reversal of exhausted Tim-3⁺PD-1⁺ NK cells, we examined the ERK/MAPK, PI3K-AKT and STAT pathways because IL-21 has been suggested to be transduced via these signalling pathways[19]. We found that IL-21 induced ERK, AKT, STAT1 and STAT3 phosphorylation (Figs 5c and 6a,c,e). When exhausted Tim-3⁺PD-1⁺ NK cells were pretreated with various signalling inhibitors, IFN-γ secretion and T-bet upregulation by IL-21 were reduced in a dose-dependent manner by Fludarabine (STAT1 inhibitor) and LY294002 (PI3K inhibitor) (Fig. 6b,f,g). However, PD98059 (ERK inhibitor) and S31-201 (STAT3 inhibitor) did not reduce IFN-γ secretion (Fig. 6d). We also analysed the Foxo1 and T-bet transcription factors. The PI3K-AKT pathway is known to enhance proliferation[32] and upregulate T-bet via Foxo1 phosphorylation[33]. The Foxo1 transcription factor negatively regulates the effector functions of NK cells by inhibiting T-bet expression[34]. We found that intratumoural rIL-21 treatment elicited Foxo1 phosphorylation and T-bet upregulation in NK cells (Supplementary Fig. 15A).

We also confirmed that the addition of exogenous rIL-21 induced Foxo1 phosphorylation in exhausted human NK cells (Supplementary Fig. 15B). Collectively, these results suggest that STAT1 and the PI3K-AKT-Foxo1 axis are crucial regulators of the IL-21-mediated reversal of exhausted Tim-3⁺PD-1⁺ NK cells.

**Reversal of exhausted NK cells in cancer patients by IL-21.** Recently, PD-1⁺ and Tim-3⁺ NK cells were identified in the tumour tissues of cancer patients[11,35,36]. However, the characteristics of these intratumoural NK cells in cancer patients have not yet been fully investigated. To confirm the presence of Tim-3⁺PD-1⁺ NK cells in cancer patients, we analysed tumour tissues isolated from patients with different types of cancer. When we isolated NK cells from human tumour tissues based on CD16 and CD56 co-expression, the expression levels of Tim-3 and PD-1 on intratumoural NK cells were much higher than those in normal tissues of the same patients (Fig. 7a,c). MHC class I expression in EpCam⁺ tumour tissues was lower than that in EpCam⁺ normal tissues (Fig. 7b), suggesting that Tim-3 and PD-1 expression on intratumoural NK cells was presumably induced by MHC class I downregulation in the tumour tissues of cancer patients.

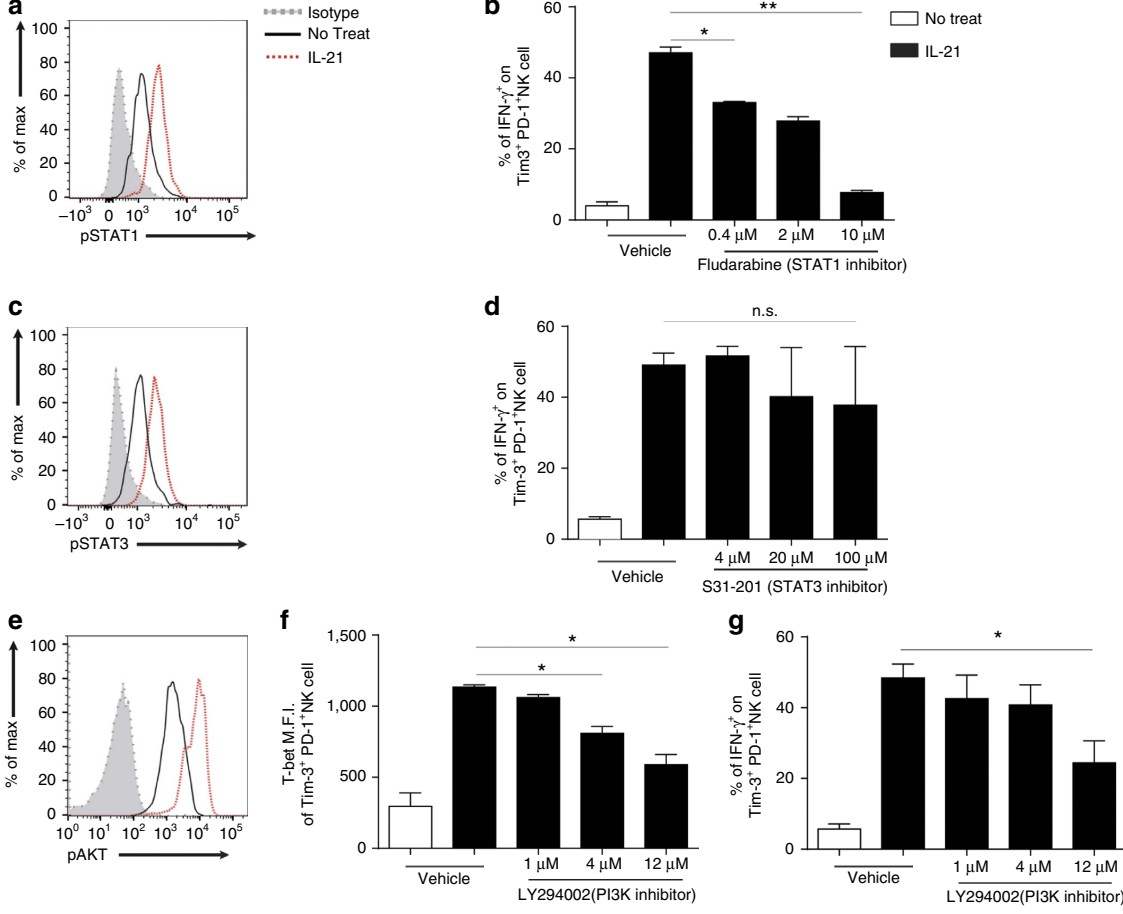

**Figure 6 | STAT1 and the PI3K-AKT pathway are associated with the IL-21-mediated functional reversal of Tim-3⁺PD-1⁺ NK cells.** Tim-3⁺PD-1⁺ NK cells were prepared from a 3-day culture of C57BL/6 splenic NK cells and H-2K^b and H-2D^b KO tumour cells. The expression of phospho-STAT1 (**a**), phospho-STAT3 (**c**) and phospho-AKT (**e**) on Tim-3⁺PD-1⁺ NK cells was analysed after a 1-h incubation with or without rIL-21 (20 ng ml⁻¹). IFN-γ (**b,d,g**) and T-bet (**f**) expression on Tim-3⁺PD-1⁺ NK cells treated with either vehicle (DMSO) or indicated concentrations of Fludarabine (STAT1 inhibitor) (**b**), S31-201 (STAT3 inhibitor) (**d**) or LY294002 (PI3K inhibitor) (**f,g**) for 30 min, followed by overnight incubation in the presence or absence of rIL-21 (20 ng ml⁻¹) and stimulation with anti-NK1.1. The data were analysed by a two-tailed unpaired Student's t-test. *P < 0.05, **P < 0.01, ***P < 0.001, ****P < 0.0001. The data are representative from at least two independent experiments. All values represent the mean ± s.e.m.

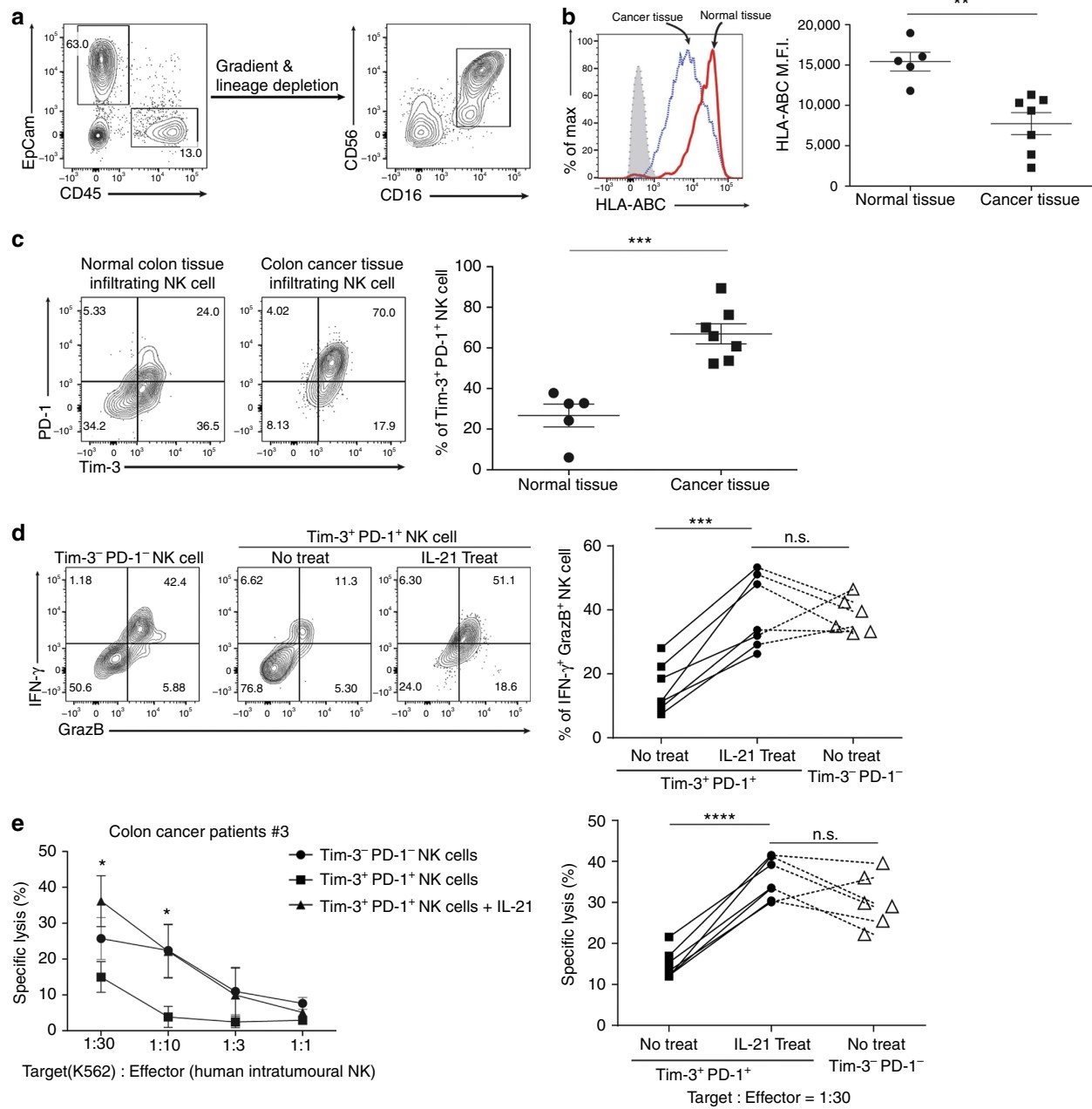

**Figure 7 | Expression level of Tim-3 and PD-1 on intratumoural NK cells in cancer patients and rescue of exhausted Tim-3$^+$ PD-1$^+$ NK cells by IL-21.** (**a**) The expression of EpCam and CD45 in the primary tumour tissues or normal tissues was analysed by flow cytometry. Intratumoural NK cells were isolated by gradient centrifugation, followed by depletion of lineage (EpCam, CD3, CD19, CD14)$^+$ cells. (**b**) HLA-ABC expression on gated EpCam$^+$ tumour or normal tissues from cancer patients. (**c**) Tim-3 and PD-1 expression on gated CD16$^+$CD56$^+$ intratumoural NK cells from cancer patients. (**d**) Isolated Tim-3$^+$PD-1$^+$ or Tim-3$^-$PD-1$^-$ NK cells were incubated overnight in the presence or absence of rIL-21 (20 ng ml$^{-1}$) and stimulated with anti-hNKp46 for 5 h. The IFN-$\gamma$ and granzyme B expression on the indicated cells is depicted. (**e**) Isolated Tim-3$^+$PD-1$^+$ or Tim-3$^-$PD-1$^-$ NK cells were incubated overnight in the presence or absence of rIL-21 (20 ng ml$^{-1}$) and were co-cultured with $^{51}$Cr-labelled K562 cells as target cells. The data were analysed by a two-tailed unpaired Student's $t$-test *$P < 0.05$, **$P < 0.01$, ***$P < 0.001$, ****$P < 0.0001$. The data are cumulative from five (normal tissue) or seven (tumour tissue) independent experiments. All values represent the mean ± s.e.m.

Functional analysis revealed that the effector function of intratumoural Tim-3$^+$PD-1$^+$ NK cells was defective as follows: the secretion of IFN-$\gamma$ and granzyme B and the cytotoxicity of the Tim-3$^+$PD-1$^+$ NK cells were significantly reduced compared with those of the Tim-3$^-$PD-1$^-$ counterparts. Intriguingly, however, rIL-21 treatment restored IFN-$\gamma$ and granzyme B secretion and the cytotoxicity of Tim-3$^+$PD-1$^+$ NK cells to levels comparable to those of Tim-3$^-$PD-1$^-$ NK cells (Fig. 7d,e, Supplementary Fig. 16). Taken together, these results suggest that

the functional exhaustion of Tim-3$^+$PD-1$^+$ NK cells in the tumour tissues of cancer patients can be restored by exogenous IL-21 treatment.

## Discussion

To date, MHC class I downregulation or deficiency has been reported in the tumour microenvironment of patients with various advanced cancers[2,23,26]. The tolerogenic tumour microenvironment induced by MHC class I insufficiency in

tumour cells is one of the major impediments to effective cancer immunotherapy due to the restriction of the cytotoxicity of T lymphocytes and the induction of NK-cell dysfunction[5]. Thus, it remains an unsolved question whether the concomitant activation of both adaptive and innate immune responses by immunotherapies is sufficient to overcome the tolerogenic tumour microenvironment found in advanced cancers containing MHC class I-deficient cells. Here, we demonstrate that both the induction of adaptive immune responses and the reversal of NK-cell exhaustion are critical for the treatment of late-stage cancers.

Previous studies have demonstrated that the activation of NK cells is regulated by the integration of signals derived from activation and inhibitory receptors expressed on their surface[37]. For example, triggering inhibitory receptors on NK cells by the ligation of MHC class I expressed on target cells interferes with cytotoxicity-promoting signals from activating receptors, resulting in the inhibition of target cell lysis by NK cells[38]. In contrast, the downregulation of MHC class I on target cells releases NK cells from inhibitory signals to induce target cell lysis[39]. Paradoxically, MHC class I deficiency in advanced human cancers has been associated with NK-cell hyporesponsiveness, which suggests that MHC class I-deficient tumour cells have evolved to escape immunosurveillance by NK cells through previously unrecognized effects on NK cells[4,5,11,23]. In this regard, our findings provide direct evidence to explain why NK cells in cancer patients are unable to kill MHC class I-deficient or low-expressing tumour cells.

Tim-3 and PD-1 have recently emerged as specific markers for T-cell exhaustion during chronic viral infection and cancer progression[7,27]. Although the functional exhaustion of NK cells has also been demonstrated in both tumour-bearing mice and cancer patients, cell surface markers that can identify exhausted NK cells in the cancer microenvironment remain unclear[10,35]. Here, we found that the co-inhibitory receptors Tim-3 and PD-1 marked functionally defective cells among tumour-infiltrating NK cells in mice and humans alike and that tumour cells, especially MHC class I-deficient tumour cells, directly induced Tim-3$^+$ PD-1$^+$ NK cells. We hypothesized that perhaps, similar to T-cell studies in which multiple rounds of TCR stimulation induce PD-1 in chronic viral infection, chronic interaction with MHC class I-deficient tumours induces these markers on NK cells. In addition, the degranulation, cytokine secretion and cytotoxicity of Tim-3$^+$ PD-1$^+$ NK cells were more defective than those of Tim-3$^-$ PD-1$^-$ NK cells. Our study also showed that the expression levels of Tim-3 and PD-1 on intratumoural NK cells from patients with different types of cancer were much higher than those in normal tissues of the same patients. We also found that the expression of Tim-3 and PD-1 on intratumoural NK cells was inversely correlated with MHC class I expression in the tissues of cancer patients. Although previous studies have demonstrated that PD-1 and Tim-3 expression on NK cells from cytomegalovirus-infected or cancer patients transmits negative signals on NK cell cytotoxicity[8,10,11,35], a few unknown features remain to be elucidated regarding the induction of the functional exhaustion of NK cells, including (i) whether the ligation of PD-1/PD-L1 or Tim-3/galactin-9 on NK cells is required for the induction of NK-cell exhaustion, (ii) the molecular mechanism by which NK-cell exhaustion is induced by tumour cells and (iii) the molecular mechanisms involved in the upregulation of PD-1 and Tim-3 on NK cells. For example, it has been demonstrated that Tim-3 or PD-1 ligation on T cells inhibits TCR stimulation-dependent signalling pathways and promotes exhaustion of CD8$^+$ T cells through the induction of NFAT[40]. Therefore, further studies are needed to identify the precise mechanism by which PD-1 and Tim-3 signalling influence the dysfunction of NK cells and to address which molecular machineries are involved in the induction of NK cell exhaustion during cancer progression.

Previous studies have demonstrated that the production of IFN-γ by NK cells increases in a STAT1-dependent manner[41,42], whereas the cytotoxicity of NK cells is abrogated through the STAT3 pathway[43], suggesting that STAT1 and STAT3 can oppose each other in the regulation of NK cells. In addition, the MAPK/ERK and PI3K/AKT pathways are important for promoting cytokine secretion and proliferation by NK cells[44,45]. Our results suggest that IL-21 treatment induces STAT1, STAT3, ERK and AKT phosphorylation. However, the functional reversal of exhausted NK cells appears to be mediated by STAT1 and the PI3K-AKT-Foxo1 pathway. These two mechanisms may act synergistically to promote the functional reversal of NK-cell exhaustion.

Prior studies have demonstrated that neo-antigens or various antigens targeted by cancer immunotherapeutic vaccines are important for inducing anti-tumour effects[46–48] and that combination therapy (checkpoint blockade, cancer vaccines, cytokines and antibodies) with the induction of adaptive and innate immune responses could eradicate large established tumours in experimental mice[49]. In agreement with these findings, our data suggest that the spontaneous downregulation of MHC class I on tumour cells contributes to tumour evasion from immunosurveillance; to our knowledge, these advanced tumours have not previously been curable by a single immunotherapy relying on endogenous immune responses, but IL-21 secretion by NKT-cell-activating vaccination or IL-21-mediated NK-cell therapies effectively eradicate these tumours by reversing NK-cell exhaustion. We suggest that not only inducing antigen-specific T cell and innate immune responses but also restoring NK-cell exhaustion via IL-21 signalling are crucial for overcoming cancer immuno-evasion. It may be important for eliminating heterogeneous tumours containing MHC class I-deficient cells, which induce NK-cell exhaustion. In addition, the provision of exogenous IL-21 may expand antigen-specific CD8$^+$ T cells and limit CD8$^+$ T-cell exhaustion[13–15,50]. In this regard, future studies should explore whether our vaccine could also restore the exhaustion of CD8$^+$ T cells in tumour microenvironments through IL-21 produced by the activated NKT cells.

To our knowledge, this is the first report to define Tim-3$^+$ PD-1$^+$ NK cells as exhausted cells induced by MHC class I-deficient tumours in mice and humans. In addition, our study explains how MHC class I-deficient tumours escape cancer immunosurveillance, despite their increased susceptibility to NK cell-mediated cell cytotoxicity during the initial stage. Furthermore, we clearly demonstrated that IL-21 can reverse the functional exhaustion of NK cells by activating both the PI3K-AKT-Foxo1 and STAT1 signalling pathways. Therefore, these findings demonstrate that NK-cell exhaustion is an attractive drug target for developing anticancer immunotherapies and show that enhancing the concentration of IL-21 in the tumour microenvironment can be highly beneficial for the treatment of advanced tumours enriched with dysfunctional NK cells.

## Methods

**Mice.** Female C57BL/6 and BALB/c mice were purchased from Charles River Laboratories (Seoul, Korea). The Jα281$^{-/-}$ mice were kindly provided by Dr Doo-Hyun Chung (Seoul National University, Seoul, Korea). The Rag 1$^{-/-}$ mice were purchased from the Jackson Laboratory (Bar Harbor, ME, USA). The IFN-γ-reporter-eYFP (GREAT) mice on a C57BL/6 background were kindly provided by Dr Richard M. Locksley (University of California, San Francisco, USA). The IL-2Rγ$^{-/-}$ Rag2$^{-/-}$ mice were purchased from Taconic (Germantown, New York, USA) All mice were used at 6–10 weeks of age and were bred and maintained in the specific pathogen-free vivarium of Seoul National

University. All animal experiments were approved by the Institutional Animal Care and Use Committee (IACUC) at Seoul National University.

**Human samples.** The human tumour tissue and normal tissue specimens from patients with colorectal, melanoma and bladder cancer were cut into small pieces at the Department of Surgery, Shinchon Severance Hospital, Yonsei University College of Medicine.

Human peripheral blood was obtained from healthy volunteers, and informed consent was granted from all donors. Peripheral blood mononuclear cells were prepared by Ficoll-Hystopaque (Sigma-Aldrich, USA) density gradient centrifugation. The collection of human samples and all human experiments were approved by the ethical committee of Seoul National University and Shinchon Severance Hospital, Yonsei University College of Medicine.

**Reagents and antibodies.** The antibodies for flow cytometry and western blot assays were purchased from Biolegend (San Diego, CA, USA), eBioscience (San Diego, CA, USA), BD Bioscience (San Jose, CA, USA) and Cell Signaling Technology (Danvers, MA, USA). Antibody information can be found in Supplementary Table 1. All microbeads (CD3ε, CD11b, CD19, CD14, B220, EpCam and Biotin) for lymphocyte depletion or selection were purchased from Miltenyi Biotec (Bergish Galdbach, Germany). The antibodies for in vivo depletion (Asialo GM-1, NK1.1 (PK136), CD4 (GK1.5), CD8 (2.43), IL-21R (4A9) and IFN-γ (XMG 1.2)) were prepared in the laboratory or purchased from Biolegend or BioXcell. The recombinant mouse IL-2, mouse IL-21, human IL-2, human IL-21, ULBP2 and human NKp46 antibodies were purchased from R&D Systems. Chemical inhibitors, including S31-201, Fludarabine, LY294002 and PD98059, were purchased from Selleckchem (Houston, TX, USA).

**Cell line generation and culture.** The TC-1, CT26, MC38, K562 and HeLa cells were purchased from ATCC. The cells were cultured in DMEM or IMEM (GIBCO) supplemented with 10% FBS (GIBCO) and 1% penicillin-streptomycin. The optimized sgRNA constructs, targeting H-2K$^b$ (Exon 3: 5′-AGCCGTCGTAGGCG-TACTGCTGG)-3′, H-2D$^b$ (Exon3: 5′−AGTCACAGCCAGACATCTGC TGG)-3′, mouse Beta 2 microglobulin (Exon 2: 5′-TCACGCCACCCACCGGAGAATGG)-3′ and human Beta 2 microglobulin (Exon 1: 5′-GAGTAGCGCGAGCA-CAGCTAAGG)-3′, and the Cas9 expression construct, pRGEN-Cas9-CMV, were obtained from ToolGen (Seoul, Korea). The TC-1 H-2K$^b$ and H-2D$^b$, MC38 H-2K$^b$ and H-2D$^b$, CT26 Beta 2 microglobulin and HeLa Beta 2 microglobulin knock out cell lines were generated by transfection with the indicated sgRNA construct and pRGEN-Cas9-CMV using Lipofectamine 2000 (Invitrogen) according to the manufacturer's protocol. MHC class I-deficient cells were sorted by a T7E1 assay with single cell selection or with a FACSARIA III (BD Bioscience). For primary cell cultures, mouse primary cells were cultured in RPMI (GIBCO) medium supplemented with 10% FBS (GIBCO) and 1% penicillin-streptomycin (Lonza). Human primary cells were cultured in X-VIVO15 medium (Lonza) that was supplemented with 1% penicillin-streptomycin (Lonza). For NK cell culture, a low dose (1∼2 ng ml$^{-1}$) of rIL-2 was added to the culture medium for cell survival. All cell lines were found to be negative for mycoplasma contamination.

**Construction of recombinant adenoviruses.** All adenoviral vectors were constructed by Cellid, Inc. (Seoul, Korea). To construct the adenoviral vector that expressed the HPV antigen E6/E7 gene in the E1 region of adenovirus, we first constructed a pShuttle-CMV vector (Agilent Technologies, CA, USA) that expressed the HPV antigen E6/E7 protein. The newly generated pShuttle-CMV-E6E7 vector was co-transformed with the pAdeasy1 adenovirus vector (Agilent Technologies, CA, USA) in which a portion of the Ad type 5 fibre was replaced by a portion of the Ad type 35 fibre (Adk35), yielding the plasmid pAdk35-E6E7. This recombinant plasmid was transfected into human embryonic kidney 293 cells to generate the Adk35-E6E7 adenovirus.

**B/MoAdk35-E6E7/αGC and B/Mo/αGC preparation.** Splenocytes were isolated from C57BL/6 or BALB/c mice. The granulocytes and RBCs were removed by Ficoll (Sigma-Aldrich) density gradient centrifugation. After the depletion of CD3ε$^+$ and DX5$^+$ cells from the cell population using anti-mouse CD3ε and anti-mouse DX5 microbeads, the CD11b$^+$ cells and B220$^+$ cells were purified using anti-mouse CD11b and anti-mouse B220 microbeads (all from Miltenyi Biotec). Isolated B cells and monocytes (1:1 ratio mixture) were transduced with the indicated adenoviral vector in a 9-min, 2,800-r.p.m. centrifugation step at room temperature, and the cells were subsequently incubated for an additional 18 h (for the B/Mo/αGC preparation this step was skipped). αGC (KRN7000, Enzo Life Science, Japan) was loaded into the prepared B cells and monocytes for NKT activation based on our previous studies[20–22,51].

**Transplant tumour and therapeutic tumour models.** Conventional subcutaneous indicated tumours were generated by s.c. injection in the right flank. Tumour growth was measured using a metric calliper 2–3 times a week. In some experiments, tumour cells were injected s.c. in the flank after being resuspended in

Matrigel with reduced matrix growth factor (BD Bioscience). For the therapeutic tumour model, $1 \times 10^4$–$1 \times 10^6$ of the indicated tumour cells were subcutaneously injected into the left flanks of the mice on day 0. After the randomized and blinded allocation of all tumour-bearing mice, the indicated vaccine or cytokine was administered at various time points. For the TC-1 lung metastasis model, $1 \times 10^5$ cells were injected i.v. into the tail vein after resuspension in 100 μl phosphate-buffered saline (PBS). Tumour formation was monitored using IVIS Spectrum microCT and Living Image (ver. 4.2) software (PerkinElmer, Cambridge, UK). An MMPSense 680 probe (PerkinElmer; 2 nmol per 150 ml in PBS) was used to facilitate tumour monitoring. For adoptive transfer into the IL-2Rγ$^{-/-}$ Rag2$^{-/-}$ mice, tumour cells were subcutaneously injected into the left flank of the mice, and on the next day, the indicated cells were transferred by i.v. injection into the tail vein. In addition, the transferred groups of No treatment, Tim-3$^-$ PD-1$^-$ NK cells and IL-21-treated Tim-3$^+$ PD-1$^+$ eYFP$^-$ NK cells were injected daily with rIL-21 (2 μg) via i.p. for 4 days. To deplete NK1.1$^+$, CD4$^+$, CD8$^+$ cells, 300 μg of GK1.5, 2.43, PK136 or control antibodies were injected i.p. twice weekly into the mice the day before tumour implantation. To block IL-21R or IFN-γ, anti-IFN-γ (500 μg per mouse) or anti-IL-21R (300 μg per mouse) or both were injected i.p. nine days after tumour implantation, and two days later, the indicated cellular vaccine was injected along with the neutralizing antibodies.

**Antibody staining and flow cytometry analysis.** Dead cells were excluded by staining with Fixable Viability Dye (eBioscience, CA, USA) following the manufacturer's instruction. The cells were stained with the specified antibodies in 50 μl of FACS buffer (PBSN + 1% FBS). Intracellular staining for cytokines was performed after surface staining with the Cytofix/Cytoperm kit (BD Bioscience). For transcription factor staining, the Foxp3/Transcription factor staining buffer set (eBioscience) was used. In addition, for the intracellular staining of p-ERK, p-AKT, p-FOXO1, pSTAT1 and pSTAT3, the cells were fixed with IC fixation buffer and permeabilized with 1 ml ice-cold 90% methanol on ice for at least 30 min in the dark after surface staining. The samples were acquired with a FACSCalibur or FACSARIA III instrument (BD Bioscience), and the data were analysed with Flow Jo software (Three Star).

**Tumour-infiltrating lymphocyte preparation.** Tumours were dissociated using the gentleMACS Dissociator (Miltenyi Biotec). The dissociated tumour samples were further digested in 10% FBS with RPMI medium containing 300 μg ml$^{-1}$ collagenase D (Roche), 20 μg ml$^{-1}$ hyaluronidase (Sigma Aldrich) and 20 μg ml$^{-1}$ DNase I (Sigma Aldrich). After filtering through a 40-μm nylon mesh, single cell suspensions were counted and used for the experiments.

**In vitro cytotoxicity assay.** Effector CD8$^+$ cells were prepared from vaccinated mouse splenocytes and stimulated with the E6$_{41-50}$ [EVYDFARDL]/E7$_{49-57}$ [RAHYNIVTF] peptide mixture for 5 days. After the 5-day stimulation, CD8$^+$ T cells were isolated using anti-mouse CD8 microbeads (Miltenyi Biotec) and co-cultured with $^{51}$Cr-labelled TC-1 tumour cells for 4 h. The CTL activity was calculated by the release of 51Cr in the culture supernatants through the specific lysis of TC-1 target cells, as measured by a Wallac 1480 Wizard automatic γ-counter (PerkinElmer). NK cells were prepared from vaccinated mouse splenocytes or infiltrating lymphocytes from MHC class I-deficient tumour-bearing mice or human infiltrating NK cells from cancer patients by FACSARIA II or FACSARIA III sorting. Next, various numbers of NK cells were co-cultured with $^{51}$Cr-labelled TC-1 H-2K$^b$ and H-2D$^b$ KO cells or K562 cells. The NK-cell cytotoxicity was calculated by $^{51}$Cr release in the culture supernatants through the specific lysis of TC-1 H-2K$^b$ and H-2D$^b$ KO target cells or K562 cells, as measured by a Wallac 1480 Wizard automatic γ-counter (PerkinElmer). To assay the cytokine release from NK cells, $3 \times 10^4$∼$1 \times 10^6$ cells were stimulated in flat-bottomed, high protein-binding plates (Corning) that were coated with anti NK1.1, ULBP-2 Fc protein or hNKp46 antibodies for 5 h in the presence of 1 μg ml$^{-1}$ GolgiPlug (BD) before staining the cells for intracellular IFN-γ, perforin or granzyme B. In some experiments, CD107a was added during cell stimulation. When NK cells were co-cultured with tumour cells, the tumour cells were irradiated (5,000 rads) using a gamma irradiator (GC 3000 Elan) at the National Center for Inter-university Research Facilities (NCIRF) at Seoul National University.

**Immunoblot analysis.** The cytoplasmic fractions of cells were prepared as follows: cells were washed once with ice-cold PBS and collected by centrifugation at 3,000 r.p.m. for 5 min. The cells were resuspended in 10 mM HEPES, pH 7.9, 10 mM KCl, 0.2 mM EDTA, 1 mM DTT, 0.25 mM PMSF, and proteinase inhibitor cocktail. After incubation on ice for 5 min, NP-40 was added to a final concentration of 0.25%. The mixtures were vortexed at high speed for 10 s. The extracts were collected by centrifugation at 13,000 r.p.m. for 1 min. The supernatants were collected as cytoplasmic extracts. The uncropped scans of the western blot were provided as Supplementary Fig. 17.

**Immunohistochemistry.** Tumour tissues were fixed in 4% paraformaldehyde; paraffin-embedded tissue sections (4 mm) were deparaffinized, dehydrated and treated with 3% hydrogen peroxide to block endogenous peroxidase. Slides were

incubated with the primary antibody (Anti-MHC class 1 H-2 D$^b$ antibody, Abcam Inc, MA, USA). Slides were then washed with PBS, incubated with HRP secondary antibody and washed again in PBS; colour was developed using the AEC substrate. The slides were counterstained with haematoxylin.

**Enzyme-linked immunosorbent assay.** The following cytokines were measured in mouse serum using an ELISA kit according to the manufacturer's instructions: IL-21 (eBioscience), IFN-γ, TNF-α and IL-4 (BD Bioscience, San Jose, CA, USA).

**Statistics.** Statistical comparisons were performed using the Prism 6.0 software (GraphPad Software). $P$ values were determined as follows:

- One-way ANOVA Kruskal–Wallis test with Dunn's multiple comparisons: Supplementary Fig. 14B, D and E. Two-way ANOVA with Bonferroni multiple comparisons test: Figs 1a,b,2a–g and 5g–i,k, Supplementary Figs 1A–C,9B,14A and 16.
- Unpaired two-tailed Student's $t$-test: Figs 3a–e and 4b–e, 5b,d,f and 6b,d,f,g, 7b–e, Supplementary Figs 6A,7C,11A,B,12B,13 and 15.
- A log-rank (Mantel–Cox) test (conservative): Figs 1c,2h–j.

**Data availability.** All data of this study are available within the article and its Supplementary Information files or from the corresponding author on reasonable request.

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

## Acknowledgements

We thank Dr R. M. Locksley (University of California, San Francisco) for the IFN-γ-reporter-eYFP (GREAT) mice, Dr D.-H. Chung (Seoul National University) for the Jα281$^{-/-}$ mice, and the members of the Kang laboratory for technical support. We acknowledge the NIH Tetramer Core Facility for providing the Mouse CD1d-PBS-57-loaded tetramers. We appreciate the staff of the National Center for Interuniversity Research Facilities (NCIRF) at Seoul National University for assistance with the cell

sorting by flow cytometry (FACSARIA II, BD Bioscience) and the cell irradiation by the gamma irradiator (GC 3000 Elan). This work was supported by a grant from the Basic Science Research Program (NRF-2015R1A2A1A10055844), Bio & Medical Technology Development Program (NRF-2016M3A9B5941426), and X-project (NRF-2016R1E1A2A01939643) through the National Research Foundation of Korea funded by the Ministry of Science, ICT & Future Planning, and the Korea Health Technology R&D Project through the Korea Health Industry Development Institute (KHIDI), funded by the Ministry of Health & Welfare, Republic of Korea (grant number: HI14C1031).

## Author contributions

H.S., B.-S.K., S.-J.K., S.J.S., Y.C., C.-Y.K. designed the study. I.J., K.C., T.O. constructed the Adk35-E6E7 viral vector. H.S., B.S.M., Y.D.H., S.J.S. performed the human sample experiments. H.S., M.P., I.J., B.S., E.-A.B., C.-H.K., K.-S.S., J.M. performed the immunological experiments. H.S. conducted the statistical analysis. H.S., B.-S.K., Y.C., C.-Y.K. analysed the data. All authors interpreted the data. H.S., E.-A.B., B.-S.K., C.-Y.K. wrote the manuscript. All authors approved the final manuscript.

## Additional information

**Competing interests:** C.-Y.K. is employed as the chief executive officer of Cellid, Inc. The remaining authors declare no competing financial interests.

