## [Peer Review File · Nature Communications]

Reviewers' comments:

Reviewer #1 (Remarks to the Author):

The authors have examined the role of NK cell exhaustion in the anti-tumor effects of a vaccine treatment and the ability of IL-21 to reverse this event.

Comments:

1. Define "highly correlated" when referring to the effect of MHC class I expression on immune cell cytotoxicity in Fig. 1I.
2. In Fig 3A it would be of interest to examine Tim-3 expression in tumor-draining lymph nodes and splenocytes as a control.
3. In Fig. 4D, is there a difference in levels of phosphor-Erk in Tim-3+ PD-1+ NK cells after the administration of a stimulus such as engagement of Fc receptors by immobilized IgG?
4. Is there evidence that IL-4 and TNF-alpha are not important NKT secreted cytokines.
5. Are there any other models of NK cell exhaustion that could be tested in a similar manner?

Reviewer #2 (Remarks to the Author):

The manuscript by Seo et al. further dissect mechanisms of action of a complex vaccine consisting of NKT cell ligand loaded tumor antigen expressing APCs (B and Mo) in different transplanted mouse tumor models. The findings are also translated to human NK cells in tumor tissues. The anti-tumor efficacy of this vaccine was elucidated by the authors in previous studies. Now the authors observe a vaccine induced gradual loss of MHC class I expression by tumor cells that correlates to a higher anti-tumor activity of splenic NK cells. Moreover, a PD-1+TIM3+ population emerged in MHC class I negative tumors that was functionally exhausted. This population was also seen in human tumors. The authors demonstrate by IL21 injection and blockade of IL21R that IL-21 produced by NKT cells mediated the anti-tumor effect of the vaccine and reverted exhaustion. This reversion of exhaustion is also observed with human NK cells in the tumor tissue.

The study is very comprehensive and employs different types of transplanted tumor mouse models and human NK cells from tumor tissue. It extends previous studies of the same group regarding the complex tumor vaccine. PD1 expression and Tim3 expression has been shown on tumor infiltrating NK cells before (citations 8,9). Also functional exhaustion of intratumoral NK cells is not novel (citation 4). The effects of IL21 injection also have been demonstrated (Smyth et al. JEM, 2005). However, the authors succeed in combining all these findings resulting in a connected pathway leading to robust anti-tumor activity.

- 1.) PD-1 expression on human NK cells has been documented by the Moretta group (Pesce et al, 2016) showing PD1 expression on subsets of NK cells. PD-1 expression and function on human NK cells is still controversial. Original FACS plots of PD-1 expression on human cells (3E) should be shown. Does PD-1 on these cells deliver a negative signal?
- 2.) Is PD-1 a functional marker on exhausted mouse NK cells? Does it transmit negative signals? Is the ligand PDL1 observed in the tumor microenvironment? Can exhaustion also be reverted by blocking PD-1/PDL1 axis (at least in vitro)?
- 3.) In the abstract (last sentence) the authors claim that the results uncover the mechanisms involved in the induction of NK cell dysfunction. Since this is not shown in the study the sentence should be deleted. The authors show that cell/cell contact is needed for PD-1 induction but no mechanism is provided. It would be important to uncover the signals involved in the PD-1

upregulation on mouse/human NK cells.

4.) Does IL21 injection revert PD-1+ TIM3 + expression on exhausted NK cells or only their exhausted phenotype? Are side effects seen after the injection of IL21? Can dysfunction also be reverted with IL15 injections that was previously FDA approved?

[Redacted]

Minor points:

In the introduction work by Caligiuri-lab (Blood. 2010 Sep 30;116(13):2286-94) should be cited showing PD-1 expression on multiple myeloma cells and its function

Fig 2D-F: Treatment with NK1.1 might also deplete NKT cells. Please provide data about the depletion efficiency of NK versus NKT cells. Please change the sentence p5, second paragraph line 6 accordingly. Similarly p9 first sentence second paragraph

The production of IL21 by NKT cells (S9) is important for the story and should be moved into the main part.

Reviewer #3 (Remarks to the Author):

This manuscript addresses an interesting and potentially clinically-relevant topic for immunotherapy, namely the adaptive resistance of tumors through MHC class I down-regulation leading to tumor-induced NK exhaustion and the role of IL-21 overcoming it. The strengths of the study are a logical and straightforward set of experiments, a proven vaccine system, a number of convincing data sets, and new insights into NK exhaustion and how IL-21 can overcome this. Weaknesses are minimal, and mostly related to reliance on a single tumor system, using knockouts to model cells with varying levels of downregulation, and the descriptive nature of the data in figure 7. Still there are no additional experiments I would ask and I find that the message of this study will likely promote both discussion in the field and additional experimentation.

Reviewer #1 (Remarks to the Author):

The authors have examined the role of NK cell exhaustion in the anti-tumor effects of a vaccine treatment and the ability of IL-21 to reverse this event.

Comments:

1. Define "highly correlated" when referring to the effect of MHC class I expression on immune cell cytotoxicity in Fig. 1I.

[Response] We appreciate the reviewer's comment. As suggested by the reviewer, we have revised the sentence in the Results section on page 5 as follows. <original>: "This trend was highly correlated with H-2K^b, D^b downregulation on tumours (Fig. 1I)."; <revised>: "The cytotoxicity of NK cells was inversely correlated with the expression of H-2K^b, D^b on tumours, whereas that of CD8⁺ T cells was positively correlated with the expression of H-2K^b, D^b (Fig. 1I)".

2. In Fig. 3A it would be of interest to examine Tim-3 expression in tumor-draining lymph nodes and splenocytes as a control.

[Response]: We appreciate the reviewer's comment and suggestion. To address this point, we have examined Tim-3 and PD-1 expression on NK cells in tumour-draining lymph nodes and splenocytes. We observed that the expression levels of Tim-3 and PD-1 on NK cells in the spleen were very low, regardless of the MHC class I expression on tumours; however, in tumour draining lymph nodes, the expression levels were slightly increased in mice bearing H-2K^b, D^b-deficient tumours compared with WT tumour-bearing mice, although this increase

was not statistically significant (Supplementary Figure 5 in the revised manuscript).

Therefore, we have added a sentence in the Results section on page 7 as follows.

(Results) The levels of Tim-3 and PD-1 expression on NK cells in the spleen were very low, regardless of MHC class I expression on tumours, but those in tumour-draining lymph nodes were slightly increased in mice bearing H-2K^b, D^b-deficient tumours compared with WT tumour-bearing mice, although the increase was not statistically significant (Supplementary Fig. 5).

Supplementary Figure 5. Seo *et al.*

3. In Fig. 4D, is there a difference in levels of phosphor-Erk in Tim-3+ PD-1+ NK cells after the administration of a stimulus such as engagement of Fc receptors by immobilized IgG?

[Response]: We appreciate the reviewer’s comment. As the reviewer suggested, we have determined whether control IgG stimulation can affect the level of phosphor-ERK in Tim-3⁺PD-1⁺ NK cells. We observed that control IgG stimulation did not increase the level of phosphor-ERK in Tim-3⁺PD-1⁺ NK cells (Fig. 4D in the revised manuscript). Therefore, we have added these results in a revised Figure 4D.

A revised Figure 4D. Seo *et al.*

4. Is there evidence that IL-4 and TNF-alpha are not important NKT secreted cytokines.

[Response]:

We appreciate the reviewer's comment. To address this point, we have examined whether TNF- α or IL-4 can restore cytokine production by exhausted NK cells. When B/Mo/ α GC was injected, we confirmed that the NKT cells produced IL4 and TNF- α (Supplementary Fig. 12A). However, our data showed that the addition of neither IL-4 nor TNF- α retrieved the IFN- γ -producing capacity of exhausted NK cells (Supplementary Fig. 12B in the revised manuscript). Therefore, we have added a sentence in the Results section on page 11 as follows.

(Results) However, the addition of IL-4 or TNF- α did not retrieve the IFN- γ -producing capacity of exhausted NK cells (Supplementary Fig. 12B)."

Supplementary Figure 12. Seo *et al.*

5. Are there any other models of NK cell exhaustion that could be tested in a similar manner?

[Response]

We appreciate the reviewer's constructive comment. Previous studies have demonstrated that virus-encoded proteins inhibit MHC class I expression on the surface of mouse/human cytomegalovirus-infected cells^{1,2}. In addition, a recent study by Pesce *et al.* (2017, *J Allergy Clin Immunol*) demonstrated that most cytomegalovirus-infected human NK cells express high levels of PD-1 and that the frequency of PD-1⁺ NK cells was increased in patients with ovarian carcinoma. PD-1⁺ NK cells displayed the phenotypic characteristics of fully mature NK cells and exhibited defects in proliferation and cytotoxicity³. These studies raise the possibility that not only the establishment of MHC class I-deficient tumours but also viral infection can induce NK cell exhaustion. Further studies are needed to verify whether the same mechanism of action as in the tumour models is involved in the viral infection-mediated exhaustion of NK cells. Therefore, we have added a sentence to address this issue in the Discussion section on page 16 as follows.

(Discussion) “Although previous studies have demonstrated that PD-1 and Tim-3 expression on NK cells from cytomegalovirus infected or cancer patients transmits negative signals on NK cell cytotoxicity^{3, 4, 5, 6}

Reviewer #2 (Remarks to the Author):

The manuscript by Seo et al. further dissect mechanisms of action of a complex vaccine consisting of NKT cell ligand loaded tumor antigen expressing APCs (B and Mo) in different transplanted mouse tumor models. The findings are also translated to human NK cells in tumor tissues. The anti-tumor efficacy of this vaccine was elucidated by the authors in previous studies. Now the authors observe a vaccine induced gradual loss of MHC class I

expression by tumor cells that correlates to a higher anti-tumor activity of splenic NK cells. Moreover, a PD-1+TIM3+ population emerged in MHC class I negative tumors that was functionally exhausted. This population was also seen in human tumors. The authors demonstrate by IL21 injection and blockade of IL21R that IL-21 produced by NKT cells mediated the anti-tumor effect of the vaccine and reverted exhaustion. This reversion of exhaustion is also observed with human NK cells in the tumor tissue.

The study is very comprehensive and employs different types of transplanted tumor mouse models and human NK cells from tumor tissue. It extends previous studies of the same group regarding the complex tumor vaccine. PD1 expression and Tim3 expression has been shown on tumor infiltrating NK cells before (citations 8,9). Also, functional exhaustion of intratumoral NK cells is not novel (citation 4). The effects of IL21 injection also have been demonstrated (Smyth et al. JEM, 2005). However, the authors succeed in combining all these findings resulting in a connected pathway leading to robust anti-tumor activity.

1.) PD-1 expression on human NK cells has been documented by the Moretta group (Pesce et al, 2016) showing PD1 expression on subsets of NK cells. PD-1 expression and function on human NK cells is still controversial. Original FACS plots of PD-1 expression on human cells (3E) should be shown. Does PD-1 on these cells deliver a negative signal?

[Response]:

We appreciate the reviewer's comment and suggestion. We have added an original FACS plot of PD-1 expression on human NK cells in a revised Figure 3E in the revised manuscript.

A revised Figure 3E. Seo *et al.*

Pesce *et al* (2017, *J Allergy Clin Immunol*) showed that the frequency of PD-1⁺ NK cells was increased in patients with ovarian carcinoma or infected with human cytomegalovirus. They also showed that human PD-1⁺ NK cells displayed low proliferative responses and functional defects, which could be partially restored by PD-1/PD-L1 signalling blockade ³. Benson Jr *et al* (2010, *Blood*) demonstrated that human NK cells from multiple myeloma patients expressed PD-1 and that the engagement of PD-1 with PD-L1 down-regulated NK cell-versus-multiple myeloma effects. They also showed that CT-011 (a novel anti-PD-1 blocking antibody) enhanced the NK cell-versus-multiple myeloma effects ⁶. These studies suggest that the PD-1/PD-L1 axis on human NK cells might transmit negative signals. Although we do not provide direct evidence that the PD-1 signal delivers negative signals on human NK cells, we have performed additional experiments in a mouse *in vivo* tumour model using an anti-PD-1 blocking Ab to determine whether PD-1 signalling in NK cells is important for the exhaustion of NK cells. We have described these results in the response to Reviewer #2's question #2.

2.) *Is PD-1 a functional marker on exhausted mouse NK cells? Does it transmit negative signals? Is the ligand PDL1 observed in the tumor microenvironment? Can exhaustion also be reverted by blocking PD-1/PDL1 axis (at least in vitro)?*

[Response]:

We appreciate the reviewer's constructive comment. To address this point, we have

examined whether tumour cells express PD-L1. Our data showed that MC38 tumour cells exhibited a very high expression of PD-L1 on their surface (Supplementary Figure 9A in the revised manuscript), as did TC-1 tumour cells (data not shown). To determine whether the PD-1/PDL-1 axis could transmit a negative signal on exhausted NK cells, we treated MC38 H-2K^b, D^b KO tumour-bearing mice with anti-PD-1 antibody. To rule out the effects of PD-1/PDL-1 signalling blockade on CD8⁺ T cell, the mice were injected with a CD8⁺ T cell depletion mAb (clone 2.43) before tumour inoculation. We found that treatment with the anti-PD-1 Ab resulted in a dramatic regression of tumour growth in mice bearing MC38 H-2K^b, D^b KO tumours (Supplementary Figure 9B in the revised manuscript). Collectively, our data indicate that the PD-1/PD-L1 interaction transmits negative signals on NK cells and that blockade of the PD-1 signalling pathway can retrieve the antitumour effects of exhausted NK cells. We have added text in the Results section on pages 9-10 of the revised manuscript describing these new data.

(Results) “To determine whether the PD-1/PD-L1 axis transmits negative signals on NK cells, we examined whether the tumour cells expressed PD-L1. Our data showed that MC38 tumour cells had a high expression level of the PD-L1 molecule on their surface (Supplementary Fig. 9A). To examine the effect of anti-PD-1 antibody on exhausted NK cells, we treated MC38 H-2K^b, D^b KO tumour-bearing mice, which were depleted of CD8⁺ T cells by 2.43 mAb, with the anti-PD-1 antibody. This treatment significantly suppressed tumour growth in mice bearing MC38 H-2K^b, D^b KO tumours (Supplementary Fig. 9B). Collectively, our data suggest that the PD-1/PD-L1 axis transmits negative signals on NK cells and that blocking the PD-1 signal could restore the antitumour effects of exhausted NK cells”

A**B**
Supplementary Figure 9. Seo *et al*

3.) In the abstract (last sentence) the authors claim that the results uncover the mechanisms involved in the induction of NK cell dysfunction. Since this is not shown in the study the sentence should be deleted. The authors show that cell/cell contact is needed for PD-1 induction but no mechanism is provided. It would be important to uncover the signals involved in the PD-1 upregulation on mouse/human NK cells.

[Response]:

We appreciate the reviewer's critical suggestion regarding this matter. We have revised the sentences in the Abstract section on page 2 according to the reviewer's suggestion

: <original>: "These results uncover the mechanism involved in the induction of NK cell dysfunction"; <revised>: "These results reveal the process involved in the induction of NK cell dysfunction."

We also appreciate and agree with the reviewer's comments concerning the importance of defining the molecular mechanisms involved in the upregulation of PD-1 on NK cells. To address this question, we are planning a separate study. Considering the complexity of the signalling pathways associated with the induction of costimulatory molecules, it would be more beneficial to investigate them extensively in a future study. We

have described the reviewer's suggestion in the Discussion on page 16 as follows; <revised> (Discussion) including and (iii) molecular mechanisms involved in the upregulation of PD-1 and Tim-3 on NK cells. For example, it has been demonstrated that Tim-3 or PD-1 ligation on T cells inhibits TCR stimulation-dependent signalling pathways and promotes the exhaustion of CD8⁺ T cells through the induction of NFAT⁷. Therefore, further studies are needed to identify the precise mechanism by which PD-1 and Tim-3 signalling influence the dysfunction of NK cells and to address which molecular machineries are involved in the induction of NK cell exhaustion during cancer progression".

4.) Does IL21 injection revert PD-1+ TIM3 + expression on exhausted NK cells or only their exhausted phenotype? Are side effects seen after the injection of IL21? Can dysfunction also be reverted with IL15 injections that was previously FDA approved?

[Response]:

We appreciate the reviewer's comment. To address this point, we have analysed whether the addition of exogenous rIL-21 can revert the expression of Tim-3 and PD-1 on exhausted NK cells *in vitro*. Our data showed that the expression of Tim-3 and PD-1 on exhausted NK cell was not reverted by the addition of rIL-21 *in vitro*, despite the significant recovery of the IFN- γ -producing capacity of exhausted NK cells by rIL-21. Based on these results, we suggest that rIL-21 can restore the function of exhausted NK cells without affecting their Tim-3 and PD-1 expression.

Exogenous rIL-21 treat on NK cell *in vitro*

Unpublished data (reference only)

We did not test the toxicity or the side effects of rIL-21 administration. The safety of rIL-21 has been validated in Phase I and Phase II clinical trials for melanoma, renal cell carcinoma and metastatic colorectal cancer^{8, 9, 10, 11, 12}. The most common adverse events include flu-like symptoms, pruritus, and rash, most of which represent an acceptable safety profile in cancer patients. However, it is still possible that IL-21 could result in dose-limiting deleterious adverse effects at high concentrations¹³.

Similar to IL-21, IL-15 is well known as a stimulator of NK cell development, proliferation and cytotoxicity¹⁴. In addition, Beldi-Ferchiou *et al.* (2016, *Oncotarget*) showed that exogenous IL-15 could reverse the dysfunction of PD-1⁺ NK cells in patients with Kaposi sarcoma¹⁵. These previous studies suggest that rIL-15 may be an alternate treatment for reverting the dysfunction of NK cells in cancer patients.

[Redacted]

Minor points:

1) In the introduction work by Caligiuri-lab (*Blood*. 2010 Sep 30;116(13):2286-94) should be cited showing PD-1 expression on multiple myeloma cells and its function.

[Response]

We appreciate the reviewer's comment. As suggested by the reviewer, we have added the reference and text in the Introduction section on page 5 as follows.

<Original> (Introduction) "However, their expression on NK cells was not well documented until two recent human studies reported PD-1 and Tim-3 expression on NK cells of cancer patients."

<revised> (Introduction) "Although previous studies demonstrated that the PD-1/PD-L1 and Tim-3/ligands of Tim-3 signalling down-modulated the cytotoxicity of NK cells against tumour cells ^{6, 20}, their expression on NK cells was not well documented until a few recent human studies reported PD-1 and Tim-3 expression on NK cells of cancer patients ^{3, 5}"

2) Fig 2D-F: Treatment with NK1.1 might also deplete NKT cells. Please provide data about the depletion efficiency of NK versus NKT cells. Please change the sentence p5, second paragraph line 6 accordingly. Similarly p9 first sentence second paragraph

[Response]

We appreciate the reviewer's comment. As suggested by the reviewer, we have added the data (Fig S4) and revised some of the text in the Results section on pages 5 and 10 as follows. <revised> (Results) We also found that treatment with anti-NK1.1 or anti-asialo GM1 significantly reversed the inhibition of tumour growth, which indicated that the antitumour immunity was dependent on NK and NKT cells (Supplementary Figure 4 and Figure 2D-F). Given that vaccination with B/Mo/αGC effectively controls the growth of MHC class I-deficient tumours in an NK- and NKT cell-dependent manner,

Supplementary Figure 4. Seo *et al.*

3) The production of IL21 by NKT cells (S9) is important for the story and should be moved into the main part.

[Response]

We appreciate the reviewer’s valuable suggestion. As suggested by the reviewer, “the production of IL-21 by NKT cells” has been moved to Figure 5A.

Reviewer #3 (Remarks to the Author):

This manuscript addresses an interesting and potentially clinically-relevant topic for immunotherapy, namely the adaptive resistance of tumors through MHC class I down-regulation leading to tumor-induced NK exhaustion and the role of IL-21 overcoming it. The strengths of the study are a logical and straight forward set of experiments, an proven vaccine system, a number of convincing data sets, and new insights into NK exhaustion and how IL-21 can overcome this. Weaknesses are minimal, and mostly related to reliance on a single tumor system, using knockouts to model cells with varying levels of downregulation, and the descriptive nature of the data in figure 7. Still there are no additional experiments I

would ask and I find that the message of this study will likely promote both discussion in the field and additional experimentation.

[Response]

We deeply appreciate the reviewer's favourable comments concerning our study. Based on the reviewer's comments, we believe and hope that our study will promote both discussion in the immunotherapy field and future studies.

Other changes

We have revised the Abstract based on the guide for submit to '*Nature Communications*':
"No more than 150 words"

<Original>

~~During cancer immunoediting, tumor cells with spontaneous loss of MHC class I expression tend to gradually increase in the tumor microenvironment by preferentially escaping from immune surveillance of cytotoxic T cells even though MHC class I-deficient tumors have been historically considered to be susceptible to NK cell dependent cytotoxicity. Recent studies demonstrated that most NK cells found in the tumor microenvironment of advanced cancers are defective, releasing the malignant MHC class I-deficient tumors from NK cell-dependent immune control. Here, we showed that an NKT cell-ligand loaded tumor-antigen expressing APC-based vaccine effectively eradicated these advanced tumors that are not curable with a single immunotherapy. In this process, we found that the co-expression of Tim-3 and PD-1 marks functionally exhausted NK cells in advanced tumors and that MHC class I downregulation in tumors was closely associated with the induction of NK cell~~

exhaustion in tumor-bearing mice as well as in cancer patients. Furthermore, the recovery of NK cell function by IL-21 was critical for the anti-tumor effect of the vaccine against these advanced tumors. These results uncover the process involved in the induction of NK cell dysfunction in advanced cancers and provide guidance to develop new strategies for cancer immunotherapy.

<Revised>

During cancer immunoediting, loss of MHC class I (MHC-I) in neoplasm contributes to the evasion of tumours from host immune system. Recent studies have demonstrated that most NK-cells that are found in advanced cancers are defective, releasing the malignant MHC-I-deficient tumours from NK-cell dependent immune control. Here, we show that an NKT-cell-ligand-loaded tumour-antigen expressing APC-based vaccine effectively eradicated these advanced tumours. During this process, we have found that the co-expression of Tim-3 and PD-1 marked functionally exhausted NK-cells in advanced tumours and that MHC-I downregulation in tumours is closely associated with the induction of NK-cell exhaustion in both tumour-bearing mice and cancer patients. Furthermore, the recovery of NK-cell functions by IL-21 is critical for the anti-tumour effects of the vaccine against these tumours. These results reveal the process involved in the induction of NK-cell dysfunction in advanced cancers and provide guidance for the development of new strategies for cancer immunotherapy.

<References>

1. Desrosiers M-P, *et al.* Epistasis between mouse Klra and major histocompatibility complex class I loci is associated with a new mechanism of natural killer cell-mediated innate resistance to cytomegalovirus infection. *Nature genetics* **37**, 593-599 (2005).
2. Lin A, Xu H, Yan W. Modulation of HLA expression in human cytomegalovirus immune evasion. *Cell Mol Immunol* **4**, 91-98 (2007).

3. Pesce S, *et al.* Identification of a Subset of Human NK Cells Expressing High Levels of PD-1 Receptor: A Phenotypic and Functional Characterization. *J Allergy Clin Immunol* **139**, 335-346. e333 (2017).
4. Beldi-Ferchiou A, *et al.* PD-1 mediates functional exhaustion of activated NK cells in patients with Kaposi sarcoma. *Oncotarget*, (2016).
5. da Silva IP, *et al.* Reversal of NK-cell exhaustion in advanced melanoma by Tim-3 blockade. *Cancer Immunol Res* **2**, 410-422 (2014).
6. Benson DM, *et al.* The PD-1/PD-L1 axis modulates the natural killer cell versus multiple myeloma effect: a therapeutic target for CT-011, a novel monoclonal anti-PD-1 antibody. *Blood* **116**, 2286-2294 (2010).
7. Martinez GJ, *et al.* The transcription factor NFAT promotes exhaustion of activated CD8+ T cells. *Immunity* **42**, 265-278 (2015).
8. Davis ID, *et al.* Clinical and biological efficacy of recombinant human interleukin-21 in patients with stage IV malignant melanoma without prior treatment: a phase IIa trial. *Clin Cancer Res* **15**, 2123-2129 (2009).
9. Thompson JA, *et al.* Phase I study of recombinant interleukin-21 in patients with metastatic melanoma and renal cell carcinoma. *Journal of Clinical Oncology* **26**, 2034-2039 (2008).
10. Davis ID, *et al.* An open-label, two-arm, phase I trial of recombinant human interleukin-21 in patients with metastatic melanoma. *Clin Cancer Res* **13**, 3630-3636 (2007).
11. Grünwald V, *et al.* A phase I study of recombinant human interleukin-21 (rIL-21) in combination with sunitinib in patients with metastatic renal cell carcinoma (RCC). *Acta Oncologica* **50**, 121-126 (2011).
12. Petrella TM, *et al.* Interleukin-21 has activity in patients with metastatic melanoma: a phase II study. *Journal of Clinical Oncology* **30**, 3396-3401 (2012).
13. Santegoets SJ, Turksma AW, Powell Jr DJ, Hooijberg E, de Gruijl TD. IL-21 in cancer immunotherapy: At the right place at the right time. *Oncoimmunology* **2**, e24522 (2013).
14. Carson WE, *et al.* Interleukin (IL) 15 is a novel cytokine that activates human natural killer cells via components of the IL-2 receptor. *Journal of Experimental Medicine* **180**, 1395-1403 (1994).
15. Beldi-Ferchiou A, *et al.* PD-1 mediates functional exhaustion of activated NK cells in patients with Kaposi sarcoma. *Oncotarget* **7**, 72961-72977 (2016).
16. Elsaesser H, Sauer K, Brooks DG. IL-21 is required to control chronic viral infection. *Science* **324**, 1569-1572 (2009).
17. Fröhlich A, *et al.* IL-21R on T cells is critical for sustained functionality and control of chronic viral infection. *Science* **324**, 1576-1580 (2009).
18. John SY, Du M, Zajac AJ. A vital role for interleukin-21 in the control of a chronic viral infection. *Science* **324**, 1572-1576 (2009).
19. Sutherland AP, Joller N, Michaud M, Liu SM, Kuchroo VK, Grusby MJ. IL-21 promotes CD8+ CTL activity via the transcription factor T-bet. *J Immunol* **190**, 3977-3984 (2013).
20. Ndhlovu LC, *et al.* Tim-3 marks human natural killer cell maturation and suppresses cell-mediated cytotoxicity. *Blood* **119**, 3734-3743 (2012).

REVIEWERS' COMMENTS:

Reviewer #1 (Remarks to the Author):

The authors have addressed all concerns

Reviewer #2 (Remarks to the Author):

All concerns were addressed.

Reviewer #3 (Remarks to the Author):

As my initial comments were minimal and supported publication of this study, I will leave it to the other reviewers to evaluate whether the author's responses to their queries are sufficient.

Point-by-Point Response

REVIEWERS' COMMENTS:

Reviewer #1 (Remarks to the Author):

The authors have addressed all concerns

[Response]

We appreciate your comment. Thank you for reviewing our manuscript

Reviewer #2 (Remarks to the Author):

All concerns were addressed.

[Response]

We appreciate your comment. Thank you for reviewing our manuscript

Reviewer #3 (Remarks to the Author):

As my initial comments were minimal and supported publication of this study, I will leave it to the other reviewers to evaluate whether the author's responses to their queries are sufficient.

[Response]

We appreciate your support with favorable comments. Other reviewers also support publication of our revised manuscript.